# Signal, noise and skill in sub-seasonal forecasts: the role of teleconnections

Alexey Yu. Karpechko[1], Amy H. Butler[2], Frederic Vitart[3]

[1]Finnish Meteorological Institute, Helsinki, 00101, Finland
[2]National Oceanic and Atmospheric Administration (NOAA) Chemical Sciences Laboratory, Boulder, CO, 80305, USA
[3]European Centre for Medium-range Weather Forecasts, Reading, RG2 9AX, UK

*Correspondence to*: Alexey Yu. Karpechko (alexey.karpechko@fmi.fi)

**Abstract.** A set of relaxation experiments with a forecast model is used to explore the influence of tropical and stratospheric
teleconnections on forecast skill, variability of forecast ensemble mean (EM) and ensemble spread (ES) in the wintertime Northern Hemisphere at sub-seasonal timescales. The influence is diagnosed by comparing the relaxation experiments, which relax the temperature and wind fields in specific regions to observed values, with the free running (control) experiment. During weeks 3-6 the tropical relaxation increases the forecast skill for sea level pressure (SLP) mostly south of 50°N but also over the North Atlantic, Northern Europe and eastern Canada. The stratospheric relaxation improves the skill
mostly in high latitudes, over Europe, and North Atlantic. Skill improvements are smaller for surface temperature and total precipitation, suggesting a smaller role of the teleconnections in their predictability. The increases in skill are generally associated with increased variability of EM, considered to represent the predictable signal, and reduced ES representing noise. However, this does not happen in all areas where the skill is increased. In high latitudes, where the stratospheric impacts are strongest, the EM variability does not increase in the stratospheric relaxation experiments consistently with
increases in skill, implying that EM does not reflect the predictable signal. We suggest that the ensemble size available in the experiments (11 members) is not always enough to make it possible to fully extract signal from noise, and that larger ensembles (20-50 members or even more depending on area and variable) would be beneficial for studies of sub-seasonal predictability associated with the teleconnections in mid- and high latitudes, including windows for forecast opportunities.

## 1 Introduction

The signal-noise model of atmospheric predictability assumes that atmospheric variability consists of two components: a signal, the component that can be predicted by a forecast model given initial and boundary conditions, and a noise, a component that cannot be predicted. The model can be expressed using variances as:

$$\sigma_{total}^2 = \sigma_{signal}^2 + \sigma_{noise}^2 \tag{1}$$

where $\sigma^2_{total}$ is the total variance, $\sigma^2_{signal}$ is the signal variance, and $\sigma^2_{noise}$ is the noise variance. As the signal to noise ratio declines with forecast lead time, the skill of weather forecasts deteriorates. In the extratropics the skill usually becomes small after 10-14 days; however, sometimes the forecasting opportunity extends beyond two weeks, i.e. to sub-seasonal timescales (Mariotti et al., 2020). According to the signal-noise model, these "windows of opportunity" are characterized by an anomalously large signal to noise ratio (e.g. Newman et al., 2003). Detecting forecasting opportunities is, however, difficult and some of the related problems are analysed in this article.

In an ensemble forecast system, the predictable signal is represented by the ensemble mean (EM), and the unpredictable noise by the ensemble spread (ES). The signal-noise model (1) rewritten for a forecast system takes the form:

$$\sigma^2_{total} = \sigma^2_{EM} + \sigma^2_{ES} \tag{2}$$

While $\sigma^2_{EM}$ is the best estimate for $\sigma^2_{signal}$ and $\sigma^2_{ES}$ is the best estimate for $\sigma^2_{noise}$, in general these are not the same things because the models have structural errors and because the ensembles have a finite size. Yet, as long as the model has skill in predicting the real world, a correspondence between signal-noise ratio estimated from the properties of the forecast ensemble and the forecast skill is expected. Consequently, one can ask whether a forecasted large signal-noise ratio (either due to an anomalously large EM or anomalously small ES) indicates an enhanced predictability and skill. Previous studies demonstrated that, at sub-seasonal timescales, ES provides little information about the skill (Barker, 1991; Whitaker and Loughe, 1998; Hopson, 2014). It has been shown that the forecasting systems are often under-dispersive, and ES does not always represent the actual uncertainty (Pegion and Sardeshmukh, 2011; Kumar et al., 2014; Pegion et al., 2017), although recent studies suggest that some state-of-art models can capture the spread-skill relationship at sub-seasonal timescales (Vitart et al., 2025). On the other hand, at seasonal timescales, the skill is found to be related to the magnitude of the EM (Tang et al., 2007), and there is evidence that the same might also be true for sub-seasonal timescales (Newman et al., 2003; Pegion and Sardeshmukh, 2011; Albers and Newman, 2019).

Forecasting opportunities in the extratropics at sub-seasonal timescales typically appear when the extratropical troposphere is dynamically linked to other components of the climate system, such as the tropical atmosphere or the wintertime stratosphere (Mariotti et al., 2020; Richter et al., 2024). Such remote links are known as "teleconnections". In particular, the forecast skill in the extratropics often increases following sudden stratospheric warmings or weak polar vortex states (Sigmond et al., 2013; Karpechko et al., 2018; Butler et al., 2019; Domeisen et al., 2020; Erner et al., 2022), periods of the strong stratospheric polar vortex (Tripathi et al., 2015; Lee et al., 2020; Rao and Garfinkel 2021) or episodes of active Madden Julian Oscillation (Vitart, 2017; Ferranti et al., 2018; Lin et al., 2019). On the other hand, not all these episodes lead to detectable impacts, and appearance of these phenomena in the initial conditions does not guarantee skilful forecasts (Karpechko et al., 2017; Kautz et al., 2020; White et al., 2019; Kolstad et al., 2022). Studies suggest that forecast models may misrepresent teleconnections both from the stratosphere (Garfinkel et al., 2025; Erner and Karpechko, 2024; Afargan-

Gerstman et al., 2024) and the tropics (Stan et al., 2022; Roberts et al., 2023; Erner and Karpechko, 2024; Vitart and Balmaseda, 2024), resulting in missed forecast opportunities.

Studies of teleconnections typically focus on the associated forecast skill, but less is known about how teleconnections affect the signal and noise, how the changes in signal and noise are captured by the forecast models and how the changes in skill are related to changes in the signal-to-noise ratio. Understanding how the signal and noise properties of the forecast are

related to skill is crucial for diagnosing forecasting opportunities in operational practices. Recently, Spaeth et al. (2024) demonstrated that the forecast spread is reduced following periods of a weak polar vortex; suggesting that, contrary to earlier studies (e.g. Barker, 1991) the spread may be a predictor of the skill at sub-seasonal scales. However, whether a reduced spread is indeed associated with increased skill has not been demonstrated.

The main goal of this article is to investigate how the atmospheric teleconnections from the stratosphere and the tropics

affect the signal and noise in the forecasts and whether there is a relationship between increased signal-noise ratio and increased skill due to teleconnections. We will show that such a relationship exists if the signal-noise ratio is sufficiently large, but that it breaks down when the signal-noise ratio becomes small, a situation typical for mid- and high-latitudes at sub-seasonal timescales. We will further argue that, in many cases, extracting teleconnection signals in mid- and high latitudes may require larger ensemble sizes than are in most datasets available for research (Vitart et al., 2017); and provide

some estimations of what ensemble sizes are required for capturing a substantial fraction of the signals associated with the stratospheric and tropical teleconnections.

The method used here to isolate the effect of the teleconnections is based on the nudging technique (Jung et al., 2010a; 2010b; Hitchcock et al., 2022). In nudging (or relaxation) simulations, a forecast model is constrained to follow a predetermined state (typically reanalysis) in one part of the atmosphere considered to be the skill source, while allowing the

rest of the atmosphere to evolve freely, so that the forecast errors associated with the evolution of the skill source are eliminated. Jung et al. (2010a) showed that the nudging of the tropics and the stratosphere increases the skill of wintertime sub-seasonal forecasts over parts of the Northern Hemisphere extratropics, confirming that the teleconnections play a significant role in extratropical weather.

The study is structured as follows. The data, methods and diagnostics are presented in Section 2. Section 3 starts by

presenting the forecast skill and its relation to signal-noise ratio for sea level pressure, followed by estimations of required ensemble size. Section 3 ends with the results for surface temperature and total precipitation. Implication of the results and conclusions are discussed in Section 4.

## 2 Data and Methods

### 2.1 Data

We use data from the experiments reported in Vitart and Balmaseda (2024) and Karpechko et al. (2024). A set of extended-range ensemble reforecasts is produced by the European Centre for Medium-range Weather Forecasts (ECMWF) model version CY47R. The atmospheric component of the model has horizontal resolution Tco319 (about 32 km) and 137 levels in the vertical. The ensembles cover the period from December 1999 to January 2019 with initialization dates on 12 December, 16 December, 19 December, 23 December, 26 December, 30 December, 2 January, 6 January and 9 January of each year. This sums up to 180 hindcast ensembles, each consisting of 11 members. In addition to the free running experiment, referred to as the control experiment (CTRL), we use two relaxation experiments initialized on the same dates as CTRL. The details of the experiments are summarized in Table 1. In the first experiment (TROP), temperature and winds in the tropics are relaxed towards corresponding fields from ECMWF's ERA-5 reanalysis (Hersbach et al., 2020) with relaxation time of 12 hr. In the second experiment (STRAT), the mid- to upper-stratospheric winds and temperatures above 50 hPa are relaxed globally toward ERA-5 with a relaxation time of 12 hr. The results are shown for sea level pressure (SLP), 2-metre temperature (T2M) and total precipitation (TP). ERA-5 reanalysis (Hersbach et al., 2020) is used for forecast verification. All data are interpolated on the 1°·1° latitude-longitude grid prior to the analysis.

**Table 1**: **Description of the relaxation experiments**

| Name | Relaxation |
|------|------------|
| CTRL | No relaxation |
| TROP | Winds, temperature and humidity are relaxed between 10°S- 10°N with tapering applied at the boundaries of the nudging and 1000hPa – 0.01hPa with relaxation time of 12 hours. |
| STRAT | Winds, temperature and humidity are relaxed between 90°S- 90°N and 50hPa – 0.01hPa with tapering starting above 70 hPa with relaxation time of 12 hours. |

### 2.2 Signal, noise and skill metrics

The hindcast anomalies $X'$ are calculated for each ensemble member $m$ and initialization time $t$ with respect to seasonally varying daily mean hindcast climatologies (1999-2019), which are functions of longitude ($i$), latitude ($j$), and lead time ($l$). The climatologies are calculated for each experiment separately. The anomalies are then averaged weekly (week 1 is days 1-7; week 2 is days 8-14 …week 6 is days 36-42) and bi-weekly (weeks 1-2 is days 1-14; week 3-4 is days 15-28, week 5-6 is days 29-42). The analysis is mostly focused on bi-weekly anomalies.

The ensemble mean anomaly ($EM$) is defined at each grid point, initialization time, and lead time:

$$EM_{i,j,t,l} = \frac{1}{M}\sum_{m=1}^{M} X'_{i,j,t,l,m} \tag{3}$$

Here $M=11$ is the number of ensemble members.

Similarly, the ensemble spread ($ES$) is defined as a (biased) standard deviation across ensemble members:

$$ES_{i,j,t,l} = \sqrt{\frac{1}{M}\sum_{m=1}^{M}(X'_{i,j,t,l,m} - EM_{i,j,t,l})^2} \tag{4}$$

The biased estimate of the standard deviation is used because it simplifies the separation of the total variance into $\sigma_{EM}^2$ and $\sigma_{ES}^2$ as in Eq. 2. $ES$ in Eq. 4 is calculated from the weekly (or bi-weekly) mean anomalies, rather than from daily anomalies as in Spaeth et al. (2024) because the separation of the total variance in Eq. 2 requires that both $EM$ and $ES$ are calculated from the same anomalies.

The variance of $EM$ ($\sigma_{EM}^2$) and mean $ES$ ($\sigma_{ES}^2$) are defined at each grid point and lead time as:

$$\sigma_{EM_{i,j,l}}^2 = \frac{1}{T}\sum_{t=1}^{T} EM_{i,j,t,l}^2 \tag{5}$$

$$\sigma_{ES_{i,j,l}}^2 = \frac{1}{T}\sum_{t=1}^{T} ES_{i,j,t,l}^2 \tag{6}$$

where $T = 180$ is the total number of forecasts. (Note that the time mean ensemble mean $EM_{i,j,l}$ is zero because we are working with anomalies.) The ratio of the variables defined by Eq. 5 and Eq. 6 gives the signal-to-noise ratio ($STN$):

$$STN_{i,j,l} = \frac{\sigma_{EM_{i,j,l}}^2}{\sigma_{ES_{i,j,l}}^2} \tag{7}$$

Two measures of skill score are calculated. First is the anomaly correlation coefficient for timeseries ($\rho$) which is calculated for each grid point and lead time with respect to ERA5 reanalysis as follows:

$$\rho_{i,j,l} = \frac{\sum_{t=1}^{T} EM_{i,j,t,l} O'_{i,j,t,l}}{\sqrt{\sum_{t=1}^{T} EM_{i,j,t,l}^2}\sqrt{\sum_{t=1}^{T} O'^2_{i,j,t,l}}} \tag{8}$$

where $O'_{i,j,t,l}$ are the reanalysis anomalies calculated in the same way as the hindcast anomalies. The second skill score, the squared error ($SE$), is calculated as a squared difference between the ensemble mean and the observations:

$$SE_{i,j,t,l} = (EM_{i,j,t,l} - O'_{i,j,t,l})^2 \qquad (9)$$

We use $SE$ as a measure of skill for individual forecasts. We also looked at the mean squared error (or root mean squared error), which is a skill metric for each grid point, and found little additional information compared to that provided by $\rho$, hence these are not shown.

Comparing the forecast skill with the signal-noise estimates may be more convenient when a "perfect model" forecast skill ($\rho_{perf}$), i.e. the skill of the model predicting itself, is used instead of $STN$. Two perfect skill metrics are used. Following Pegion and Sardeshmukh (2011) we define $\rho_{perf}^{STN}$ using the $STN$ estimate for each grid point ($i,j$) and lead time ($l$) as:

$$\rho_{perf_{i,j,l}}^{STN} = \frac{STN_{i,j,l}}{\sqrt{(STN_{i,j,l}+1)(STN_{i,j,l}+\frac{1}{M})}} \qquad (10)$$

The other metric is the anomaly correlation coefficient (ACC) analogous to that defined by Eq. 8 except that the reanalysis anomalies are replaced with one forecast ensemble member and the ensemble mean is defined using the remaining $M-1$ members. Here we obtain the perfect skill metric $\rho_{perf}^{ACC}$ for each grid point and lead time by first calculating the correlation coefficients for each ensemble member and then averaging the results across all members.

## 2.3 Estimating teleconnection signals and size of ensembles

Separating signal from noise using ensemble technique assumes that the noise is cancelled out when the forecasts are averaged across ensemble members. However complete cancellation is only possible for ensembles of infinite size. For finite ensembles, $\sigma_{EM}^2$ contains uncancelled noise (e.g. Siegerd et al., 2016; Hardiman et al., 2022):

$$\sigma_{EM}^2 = \sigma_{signal}^2 + \frac{\sigma_{noise}^2}{M} \qquad (11)$$

Assuming that the model captures the signal and noise well, one can use Eq. 11 to estimate the size of ensembles required to separate the signal from noise, or more specifically, for $\sigma_{EM}^2$ to be a reasonable approximation to $\sigma_{signal}^2$. If the total variability $\sigma_{total}^2$ can be estimated from the forecast ($\sigma_{total}^2 = \sigma_{signal}^2 + \sigma_{noise}^2 = \sigma_{EM}^2 + \sigma_{ES}^2$) then expressing $\sigma_{noise}^2$ as the difference between $\sigma_{total}^2$ and $\sigma_{signal}^2$ allows one to rewrite Eq. 11 as:

$$\sigma_{signal}^2 = \frac{M \cdot \sigma_{EM}^2 - \sigma_{total}^2}{M-1}$$

The above equation can be used to estimate the strength of the signal using the forecast outputs. In this paper we are interested in the signals associated with teleconnections ($\sigma^2_{telecon}$). The effects of the teleconnections are estimated as the difference between CTRL and respective relaxation experiments. Thus, the teleconnection signal can be estimated as:

$$\sigma^2_{telecon} = \frac{M \cdot \Delta\sigma^2_{EM} - \Delta\sigma^2_{total}}{M-1} \tag{12}$$

where $\sigma^2_{telecon} \equiv \Delta\sigma^2_{signal}$, and $\Delta\sigma^2_{EM}$ and $\Delta\sigma^2_{total}$ are calculated as the differences between the respective relaxation experiment (either TROP or STRAT) and CTRL. Note that if a part of the teleconnection signal is captured by CTRL, then Eq. 12 would underestimate the magnitude of the signal. More details on using Eq. 12 are given Section 3.4.

If the teleconnection signals are expected to be well represented in free running forecasts, then the size of the ensembles should be such that $\sigma^2_{telecon} \gg \frac{\sigma^2_{noise}}{M_{req}}$, where $M_{req}$ is the minimum size of ensembles required to satisfy the inequality. In this article we arbitrarily define $M_{req}$ as the size of the ensembles required for $\sigma^2_{telecon}$ to constitute 2/3 of $\sigma^2_{EM}$ (equivalent to $\sigma^2_{telecon}$ being twice as large as $\frac{\sigma^2_{noise}}{M_{req}}$). Assuming that at sub-seasonal timescales $\sigma^2_{telecon}$ is the main contributor to $\sigma^2_{signal}$, and consequently that $\sigma^2_{noise}$ can be expressed as the difference between $\sigma^2_{total}$ and $\sigma^2_{telecon}$, the equation for $M_{req}$ is derived from Eq. 11 as:

$$M_{req} = 2 \cdot \frac{\sigma^2_{total} - \sigma^2_{telecon}}{\sigma^2_{telecon}} \tag{13}$$

Where $\sigma^2_{total}$ is the sum of $\sigma^2_{EM}$ and $\sigma^2_{ES}$ in CTRL, and $\sigma^2_{telecon}$ is calculated using Eq. 12.

## 2.4 NAO and PNA

To analyse regional predictability, we calculate North Atlantic Oscillation (NAO) and Pacific North American (PNA) indexes. The NAO index is calculated by projecting observed and forecasted SLP anomalies on the NAO loading pattern, which is calculated as the first EOF of the regional (90°W-40°E, 20°N-80°N) November to March monthly mean SLP anomalies from ERA-5. The PNA index is usually calculated from the 500-hPa geopotential anomalies but for consistency with NAO, here it is also calculated from SLP anomalies. To calculate the PNA loading pattern, we regressed November to March monthly mean SLP anomalies from ERA5 onto the monthly PNA index downloaded from the NOAA Climate Prediction Center ([www.cpc.ncep.noaa.gov](www.cpc.ncep.noaa.gov)). The loading PNA pattern is defined as the regression pattern over 210°W-90°W and 40°N-75°N. The PNA index is then calculated by projecting ERA5 and forecasted SLP anomalies on the PNA loading pattern. The original PNA index downloaded from the Climate Prediction Center and the PNA index calculated from ERA5 SLP anomalies here correlate at 0.84, which sufficiently captures the regional variability attributable to PNA.

**2.5 Statistical significance testing**

To test whether the relaxation experiments are significantly different from CTRL we perform bootstrap testing. This is done by (i) randomly selecting with replacement 180 hindcasts from the original set, (ii) recalculating the diagnostics for this reconstructed set of hindcasts, and (iii) repeating steps (i) and (ii) 1000 times. The actual difference between the experiments is considered statistically significant if it lies outside the 5%-95% range across the bootstrapped differences.

# 3 Results

**3.1 Forecast skill in SLP**

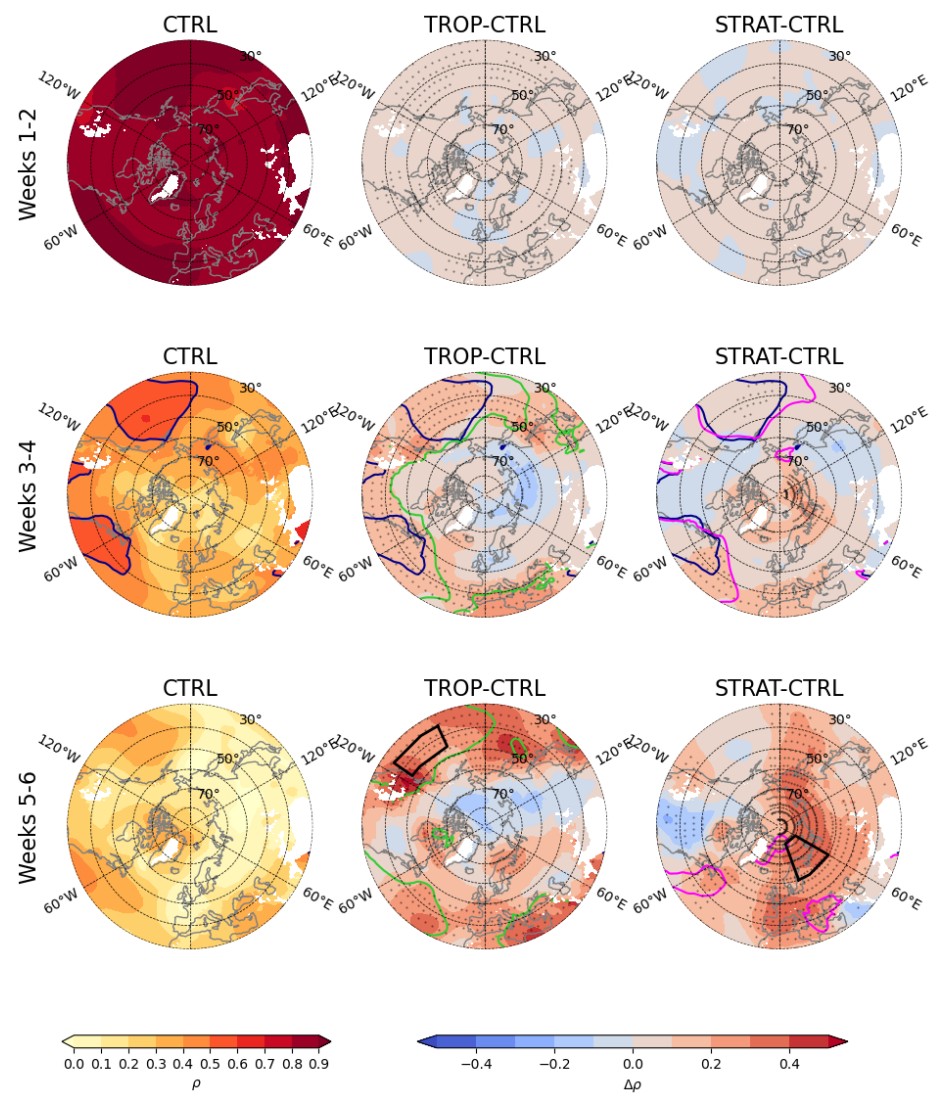

**Figure 1: Anomaly correlation coefficients ($\rho$) for bi-weekly mean SLP anomalies for (left) CTRL; and the differences in $\rho$ between (centre) TROP and CTRL and (right) STRAT and CTRL experiments. Dark blue line marks $\rho$=0.5 contour in CTRL. Light green and magenta lines mark $\rho$=0.5 contour in TROP (centre) and STRAT (right) experiments respectively. During weeks 1-2 $\rho$ exceeds 0.5 everywhere and in all experiments. During weeks 5-6 $\rho$ is below 0.5 everywhere in CTRL. Grid points with elevation above 2000 m are masked in white. The black boxes show regions used in Fig. 7. Stippling in (centre, right) indicates significant differences between CTRL and the respective relaxation experiments.**

The correlation skill score ($\rho$) for SLP fields is shown in Fig. 1. During weeks 1-2, the model has high skill, and the improvements in the relaxation experiments are small. During weeks 3-4, the skill in the mid- and high latitudes decreases considerably but remains significant ($\rho$ >0.5) in the sub-tropical Atlantic and Pacific basins. During weeks 5-6, the skill is low everywhere. The relaxation experiments reveal areas where teleconnections contribute to the skill. In TROP the skill is mostly improved south of 50°N while in STRAT the improvements are largest in high latitudes between 60°W and 180°E, in the subtropical Atlantic and western Mediterranean. The increases in skill in the Atlantic sector are expected because downward stratosphere-troposphere coupling preferentially influences this region via the North Atlantic Oscillation (e.g. Simpson and Hitchcock, 2014). In both TROP and STRAT significant skill ($\rho$ >0.5) appears over sub-tropics and extends north to 50°N-60°N over the oceans during weeks 3-4. The improvements in the relaxation experiments are in general larger in magnitude during weeks 5-6 than during weeks 3-4 because some of the signal associated with the teleconnections is captured during weeks 3-4 by CTRL.

Significant decreases of the skill can also be seen in some areas in the relaxation experiments, specifically, in northern Eurasia in TROP in weeks 3-4 and in central North America in STRAT in weeks 5-6. The decreases can indicate biases in simulated teleconnections although a more detailed analysis would be needed to test this hypothesis.

The results shown in Fig. 1 are broadly consistent with those by Jung et al. (2010), but some differences are to be expected. Firstly, the model used here has undergone significant improvements resulting in increased skill (Vitart, 2014); thus, some of the skill that could be achieved in the older model version only using the relaxation is now achieved by the free running model. Therefore, the difference between CTRL and the relaxation runs is expected to be smaller in our experiments. Secondly, we have considerably larger statistics and thus more robust estimates of the differences. Interestingly, a decrease in skill can also be seen in northern Eurasia in the older tropical relaxation experiment (Jung et al., 2010), suggesting that some biases in teleconnections might have persisted through model generations.

To highlight the regional changes, we show the correlation skill for weekly NAO and PNA indexes (Figure 2). NAO and PNA indexes provide an integral picture of the atmospheric flow over the North Atlantic /European and North Pacific /North American regions respectively. In CTRL, the NAO skill is 0.47 for week 3 but declines to ~0.2 for longer lead times. In STRAT, NAO skill is 0.52 for week 3 and remains above 0.4 for longer lead times suggesting considerable influence from the stratosphere-troposphere coupling as expected. Charlton-Perez (2021), using a statistical model, estimated that for an 11-member forecast model with a perfect stratosphere the NAO week 3 skill is ~0.52, consistent with what we find in our

experiments. In TROP the increase in the NAO skill is smaller than that in STRAT but still significantly different from CTRL during weeks 4 and 6. Karpechko et al. (2024) using the same dataset suggested that both direct tropospheric pathway (Barnes et al., 2019) and indirect stratospheric pathway contribute to increased NAO skill in TROP.

The PNA has a high skill ($\rho$=0.65) during week 3 in CTRL, indicating that the predictability in the Pacific region at sub-seasonal timescales is considerably larger compared to that over the North Atlantic. As expected, nudging the tropics has a stronger impact on PNA skill than nudging the stratosphere, and the skill in TROP remains above 0.5 even for week 5. STRAT also has a skill exceeding that in CTRL, but the difference is only significant during weeks 2 and 4. Since the zonal mean downward coupling following Sudden Stratospheric Warmings has usually small impacts over the Pacific region (Dai et al., 2023) it is not clear which processes contribute to the increased PNA skill. One possible explanation is that STRAT better predicts the wave reflection episodes which particularly affect North America (e.g. Matthias and Kretschmer, 2020), and might potentially project onto the PNA, but this hypothesis will not be tested here.

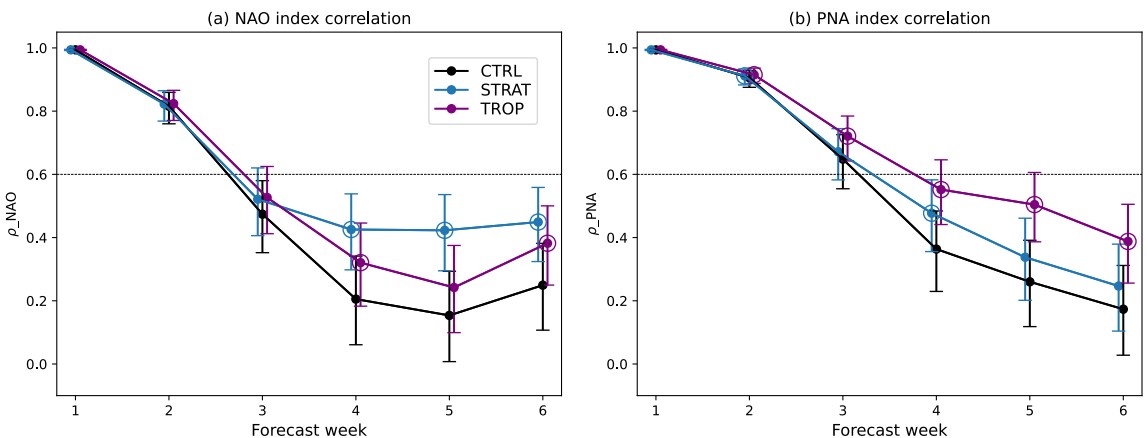

Figure 2: Anomaly correlation coefficients ($\rho$) for (a) weekly mean NAO and (b) weekly mean PNA indexes. Error bars indicate 95% confidence intervals. The coefficients for the relaxation experiments that are significantly different from those for CTRL (p=0.05) are circled.

## 3.2 Teleconnection signal and noise in SLP and the link to global skill

Signal to noise ratio (*STN*) for SLP is shown in Fig. 3. At sub-seasonal time scales (weeks 3-4 and 5-6), *STN* is largest in subtropical ocean basins – the Pacific and western Atlantic - broadly coinciding with the regions of the largest correlation skill (Fig. 1), consistent with other studies (e.g. Kumar et al., 2014). A resemblance between changes in *STN* and $\rho$ can also be seen, most clearly in TROP in the subtropics where both *STN* and $\rho$ increase, suggesting that an improved representation of the remote signal due to nudging leads, via teleconnections, to increased *STN* and consequently to increased skill. However, there are discrepancies too. For example, *STN* increases over the polar regions in TROP during weeks 3-4 (Fig. 3),

but the skill there does not change (Fig. 1). Conversely, the skill increases significantly over northern Europe in STRAT during weeks 5-6, but the changes in *STN* are minimal.


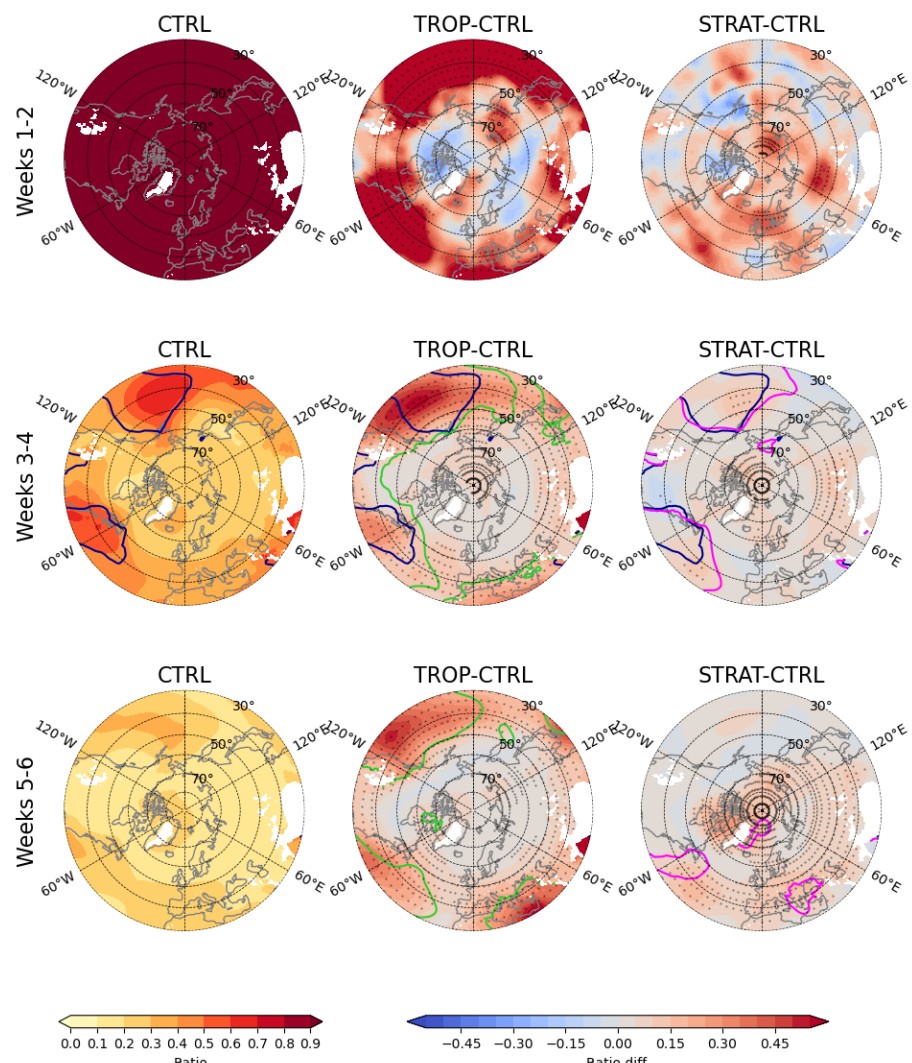

**Figure 3: Signal to noise ratio (STN) for bi-weekly mean SLP anomalies for (left) CTRL; and the differences between STN in (centre) TROP and (right) STRAT experiments with respect to CTRL. Dark blue line marks $\rho$ =0.5 contour in CTRL. Light green and magenta lines mark $\rho$ =0.5 contour in TROP (centre) and STRAT (right) experiments respectively.**


The relationship between the model's actual skill ($\rho$) and the "perfect" skill, $\rho_{perf}^{STN}$ calculated using Eq. 10 at each grid point is shown in Fig. 4. Since $\rho_{perf}^{STN}$ is a function of *STN* only, a correspondence between $\rho_{perf}^{STN}$ and $\rho$ is similar to that between

*STN* and $\rho$ (Table 2). The spatial correlation between the $\rho$ and $\rho_{perf}^{STN}$ fields in CTRL (Fig. 4a,d,g) declines with lead time,
but it remains significant until weeks 5-6. The relationship breaks down below the level of $\rho \approx 0.4$ where the actual skill
      degrades much faster than the perfect skill (Fig. 4a,d,g). The results obtained with $\rho_{perf}^{ACC}$ are very similar except that $\rho_{perf}^{ACC}$
      are smaller on average than the corresponding $\rho_{perf}^{STN}$ values, especially at sub-seasonal timescales (Fig. S1). Furthermore,
      $\rho_{perf}^{STN}$ is mostly larger than the actual skill, i.e., the values fall below the diagonal line in Fig. 4a,d,g, especially where the
      actual skill is low. However, $\rho_{perf}^{ACC}$ is more comparable to the actual skill in magnitude, and the values in Fig.S1a,d,g are
more evenly distributed around the diagonal line.

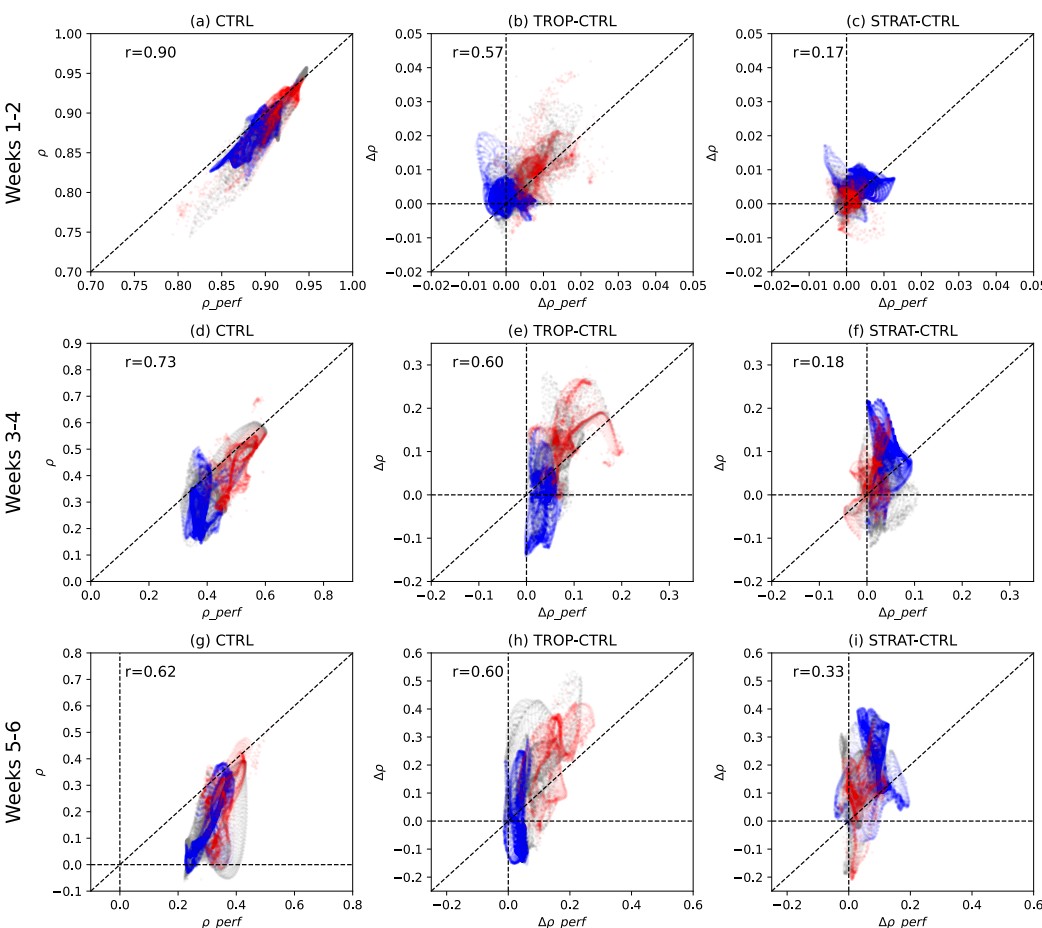

**Figure 4: (a,d,g) Scatterplots between $\rho$ and $\rho_{perf}^{STN}$ for SLP anomalies at each grid point between 30°N and 90°N in CTRL. 2nd
and 3rd columns show differences for $\rho$ and $\rho_{perf}^{STN}$ between relaxation experiments (TROP and STRAT respectively) and CTRL.
Spatial correlation coefficients between $\rho$ and $\rho_{perf}^{STN}$ fields are shown as r-values in each panel. Red dots mark grid points south of
40°N, blue dots mark grid points north of 60°N, and grey dots mark grid points between 40°N-60°N.**

**Table 2: Spatial correlation coefficients (30°N-90°N) between the forecast skill ($\rho$) and the forecast properties ($\sigma_{EM}^2$, $\sigma_{ES}^2$, STN, $\rho_{perf}^{STN}$, $\rho_{perf}^{ACC}$) in CTRL. The values above the diagonal are for Weeks 3-4. The values below the diagonal are for Weeks 5-6.**

| | $\rho$ | $\rho_{perf}^{STN}$ | $\rho_{perf}^{ACC}$ | STN | $\sigma_{EM}^2$ | $\sigma_{ES}^2$ |
|---|---|---|---|---|---|---|
| $\rho$ | 1 | 0.73 | 0.73 | 0.74 | 0.15 | -0.27 |
| $\rho_{perf}^{STN}$ | 0.62 | 1 | 1 | 0.99 | 0.26 | -0.22 |
| $\rho_{perf}^{ACC}$ | 0.62 | 1 | 1 | 0.98 | 0.26 | -0.22 |
| STN | 0.62 | 0.99 | 0.99 | 1 | 0.30 | -0.19 |
| $\sigma_{EM}^2$ | 0.10 | 0.34 | 0.33 | 0.37 | 1 | 0.83 |
| $\sigma_{ES}^2$ | -0.14 | -0.07 | -0.08 | -0.03 | 0.87 | 1 |


The relationship between the $\rho$ and $\rho_{perf}^{STN}$ fields (as well as between $\rho$ and $\rho_{perf}^{ACC}$ fields) in TROP and in STRAT behave similarly to that in CTRL (not shown). It is more interesting to focus on the changes in $\rho$ ($\Delta\rho$) and in $\rho_{perf}$ ($\Delta\rho_{perf}$) in the relaxation experiments. Since the results obtained with either $\Delta\rho_{perf}^{STN}$ or $\Delta\rho_{perf}^{ACC}$ are nearly identical (cf. Fig. 4 and Fig. S1), we only discuss the results for $\Delta\rho_{perf}^{STN}$. In TROP $\Delta\rho$ and $\Delta\rho_{perf}^{STN}$ correlate at all lead times (r~0.6), although the changes

during Weeks 1-2 are less than 0.01, i.e. negligible (Fig. 4, centre and right columns). At the subseasonal scales (Weeks 3-4, 5-6) $\Delta\rho_{perf}^{STN}$ in TROP are mostly positive, but they are often smaller than $\Delta\rho$. However, the correlation between the changes in $\rho$ and $\rho_{perf}^{STN}$ in STRAT is weak. The difference between TROP and STRAT can be understood by analysing the geographical distribution of the changes. In TROP, the increases in $\rho$ and in $\rho_{perf}^{STN}$ are mostly in low latitudes where the skill and STN are relatively large already in CTRL; while in STRAT, the increases are mostly in high latitudes, where the skill

and STN are low (Fig. 4).

Since the perfect skill is a property of a model only, in general, there need not be a correspondence between the actual and the perfect skill (Kumar et al., 2014). However, the lack of correspondence between the changes in the actual and the perfect skill in the relaxation experiments indicate that the signal-noise ratio estimated from the model does not reflect the signal and noise properties of the actual atmosphere.

We next take a closer look at $\sigma_{EM}^2$, $\sigma_{ES}^2$ and their changes in the relaxation experiments (Figs. 5-6). In general, $\sigma_{EM}^2$ is largest during weeks 1-2 and it decreases with lead time (Fig. 5). The opposite is true for $\sigma_{ES}^2$ (Fig. 6). The three regions where $\sigma_{EM}^2$ maximizes are the climatological Icelandic and Aleutian lows coincident with the location of the Northern Hemispheric storm tracks (Chang et al., 2002), and the Ural high, and these are the same regions where $\sigma_{ES}^2$ maximizes too. Overall $\sigma_{EM}^2$ and $\sigma_{ES}^2$ fields correlate strongly positively with each other at all lead times but neither of them correlates with $\rho_{perf}^{STN}$, $\rho_{perf}^{ACC}$,

or STN (Table 2). Thus, it is STN that affects $\rho$, not $\sigma_{EM}^2$ or $\sigma_{ES}^2$ separately.

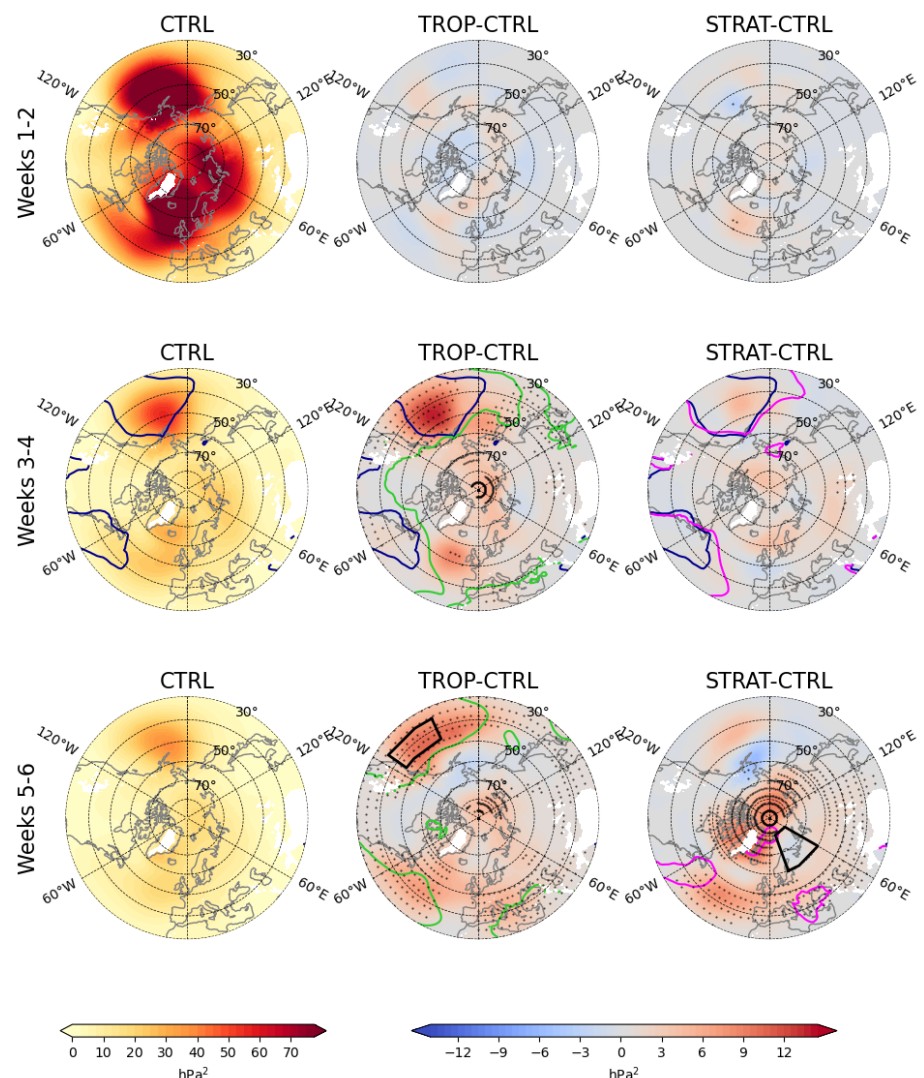

**Figure 5: Variance of ensemble mean ($\sigma^2_{\mathrm{EM}}$) for bi-weekly mean SLP anomalies for (left) CTRL; and the differences between $\sigma^2_{\mathrm{EM}}$ in (centre) TROP and (right) STRAT experiments with respect to CTRL. Dark blue line marks $\rho$=0.5 contour in CTRL. Light green and magenta lines mark $\rho$=0.5 contour in TROP (centre) and STRAT (right) experiments respectively.**


In TROP, $\sigma^2_{EM}$ mostly increases in the subtropical regions, eastern subtropical Pacific and western subtropical Atlantic where the tropical teleconnections are expected to be strong. These are also the regions where $\sigma^2_{ES}$ decreases. Also, $\sigma^2_{EM}$ increases in the Arctic but $\sigma^2_{ES}$ does not significantly change there during weeks 3-6. Recalling that there is a significant increase in $STN$ in the Arctic during weeks 3-4, we can associate this increase with increased $\sigma^2_{EM}$. In STRAT, $\sigma^2_{EM}$ increases in the Arctic,

with largest increase over northeastern Canada during weeks 5-6, as well as in the subtropical Atlantic. The largest decrease in $\sigma^2_{ES}$ is over northern Europe during weeks 5-6 but a corresponding increase in $\sigma^2_{EM}$ there is missing. Comparing now

changes in $\sigma_{EM}^2$ and $\sigma_{ES}^2$ with those of $\rho$ (Fig. 1) we see that in TROP the largest changes are in the subtropics where the relaxation increases $\sigma_{EM}^2$, decreases $\sigma_{ES}^2$ and increases $\rho$. However, in the Arctic, where increased $\sigma_{EM}^2$ contribute to increased $STN$ there is no significant increase in $\rho$. In STRAT the areas where changes in both $\sigma_{EM}^2$ and $\sigma_{ES}^2$ correspond to increased $\rho$

include Southern Europe, Mediterranean and western subtropical Atlantic during weeks 5-6. On the other hand, the increase in $\rho$ over northern Europe during weeks 5-6 is only associated with decreased $\sigma_{ES}^2$.

**Table 3: Spatial correlation coefficients over 30°N-90°N between the changes in the forecast skill ($\Delta\rho$) and changes in the forecast properties ($\Delta\sigma_{EM}^2$, $\Delta\sigma_{ES}^2$, $\Delta STN$, $\Delta\rho_{perf}^{STN}$, $\Delta\rho_{perf}^{ACC}$) in the relaxation experiments.**

| | TROP | | | | | STRAT | | | | |
|---|---|---|---|---|---|---|---|---|---|---|
| | $\Delta\sigma_{EM}^2$ | $\Delta\sigma_{ES}^2$ | $\Delta STN$ | $\Delta\rho_{perf}^{STN}$ | $\Delta\rho_{perf}^{ACC}$ | $\Delta\sigma_{EM}^2$ | $\Delta\sigma_{ES}^2$ | $\Delta STN$ | $\Delta\rho_{perf}^{STN}$ | $\Delta\rho_{perf}^{ACC}$ |
| Weeks 1-2 | 0.05 | -0.26 | 0.35 | 0.57 | 0.57 | -0.01 | -0.29 | 0.10 | 0.17 | 0.17 |
| Weeks 3-4 | 0.13 | -0.33 | 0.54 | 0.60 | 0.59 | 0.18 | -0.43 | 0.23 | 0.18 | 0.16 |
| Weeks 5-6 | 0.29 | -0.48 | 0.57 | 0.60 | 0.59 | 0.23 | -0.55 | 0.26 | 0.33 | 0.36 |


To summarize the above discussion, Table 3 lists the spatial correlation coefficients between the changes in the properties of the forecast ensemble ($\Delta\sigma_{EM}^2$, $\Delta\sigma_{ES}^2$, $\Delta STN$, $\Delta\rho_{perf}$) and the changes in the actual forecast skill ($\Delta\rho$). Looking at changes in TROP, both $\Delta\sigma_{EM}^2$ and $\Delta\sigma_{ES}^2$ perform worse than $\Delta\rho_{perf}$ (either $\Delta\rho_{perf}^{STN}$ or $\Delta\rho_{perf}^{ACC}$) in predicting $\Delta\rho$. Here, the situation is similar to what is found in CTRL where $\rho_{perf}$ correlates with $\rho$ better than either $\sigma_{EM}^2$ or $\sigma_{ES}^2$ (Table 2). Looking at changes

in STRAT, change in neither forecast property correlates well with changes in $\rho$; however, it is noticeable that $\Delta\sigma_{ES}^2$ performs better than $\Delta\rho_{perf}^{STN}$, $\Delta\rho_{perf}^{ACC}$ or $\Delta STN$. In particular, this reflects the situation over northern Europe, where increases in $\Delta\rho$ and decreases in $\Delta\sigma_{ES}^2$ coincide with each other but increase in $STN$ is small and insignificant while increase in $\sigma_{EM}^2$ is absent. This latter case suggests that the skill increase in this region might be due to reduced forecast uncertainty rather than due to increased predictable signal, and we will consider this situation in the next section.

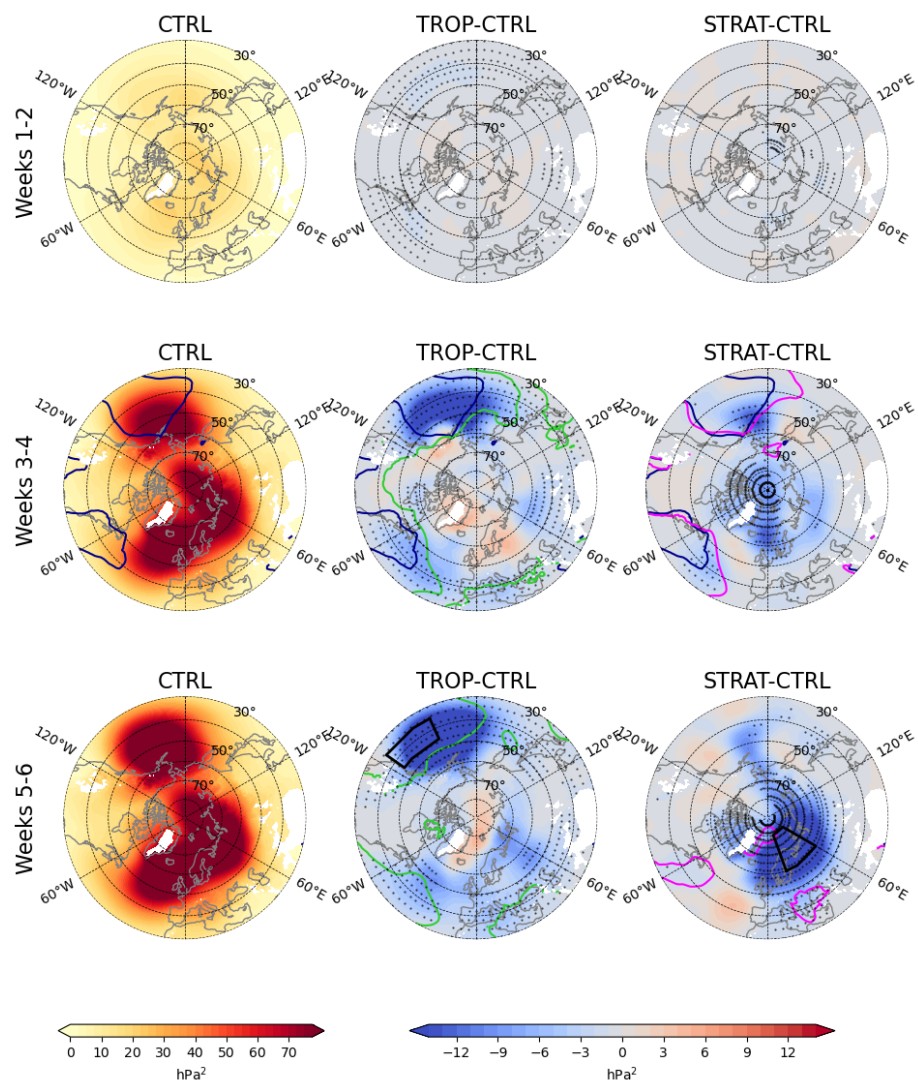


**Figure 6: Ensemble spread ($\sigma_{ES}^2$) for bi-weekly mean SLP anomalies for (left) CTRL; and the differences between $\sigma_{ES}^2$ in (centre) TROP and (right) STRAT experiments with respect to CTRL. Dark blue line marks $\rho$=0.5 contour in CTRL. Light green and magenta lines mark $\rho$ =0.5 contour in TROP (centre) and STRAT (right) experiments respectively.**

### 3.3 Teleconnection signal and noise in SLP and the link to local skill

In this section we will analyse the differences between the regions where the skill increases together with increases in $\sigma_{EM}^2$ and *STN*, as expected, and those where the skill increases, but $\sigma_{EM}^2$ and *STN* do not (or increase only slightly). The two regions considered are shown in Fig. 1. The first region is in the Eastern Pacific (35°N-45°N and 155°W-125°W) and the other one is in northern Europe (60°N-80°N and 20°E-60°E). There is a strong increase in the correlation skill relative to CTRL in the Pacific location in TROP and in the European location in STRAT during weeks 5-6 (Table 4). There is also a

significant reduction in the mean ensemble spread ($\sigma_{ES}^2$) at both locations in the respective experiments. The reduction in $\sigma_{ES}^2$ suggests that the teleconnections (the tropical and the stratospheric ones respectively) play a role in the climate variability, and hence contribute to predictability, at these locations. However, $\sigma_{EM}^2$ increases only at the Pacific location but not at the European location.

**Table 4: Forecast actual skill ($\rho$), perfect skill ($\rho_{perf}^{STN}$ and $\rho_{perf}^{ACC}$), variability in EM ($\sigma_{EM}^2$, hPa$^2$) and ES ($\sigma_{ES}^2$, hPa$^2$), STN, correlation between the changes in the forecast error (SE) and the absolute squared EM ($r_{EM^2}$), and between the SE and the ensemble spread ($r_{ES^2}$) in the two locations shown in Fig. 1 during Weeks 5-6.**

| | $\rho$ | $\rho_{perf}^{STN}$ | $\rho_{perf}^{ACC}$ | $\sigma_{EM}^2$ | $\sigma_{ES}^2$ | STN | $r_{EM^2}$ | $r_{ES^2}$ |
|---|---|---|---|---|---|---|---|---|
| | | | | Pacific | | | | |
| CTRL | 0.37 | 0.42 | 0.32 | 12 | 41 | 0.29 | -0.48 | 0.02 |
| TROP | 0.63 | 0.63 | 0.57 | 19 | 24 | 0.79 | | |
| | | | | Northern Europe | | | | |
| CTRL | 0.05 | 0.26 | 0.07 | 10 | 81 | 0.13 | -0.30 | 0.04 |
| STRAT | 0.42 | 0.30 | 0.13 | 10 | 61 | 0.16 | | |

Figure 7 shows the changes in $EM^2$, $ES^2$, and $SE$ at these locations for each individual forecast. To facilitate the analysis, the forecasts are ranked according to the magnitude of the $EM^2$ change in the relaxation experiments (Fig. 7a, c). At both locations $EM^2$ can either decrease or increase by Weeks 5-6 for each individual forecast initialization. However, in the Pacific the increases in $EM^2$ dominate over the decreases leading to an overall increase in $\sigma_{EM}^2$ and consequently in $STN$. In Europe, there is nearly as many forecasts with increased $EM^2$ as with decreased $EM^2$, which balance each other and result in 370 an overall lack of change in $\sigma_{EM}^2$.

Figure 7 (b,d) and Table 4 also show that it is the shift in the ensemble mean that leads to reduced forecast error in the individual forecasts at these locations. This is seen from a significant negative correlation between the absolute changes in $EM^2$ and the forecast error $SE$ at both locations. The absolute changes in $EM^2$ are considered because both the negative and the positive changes in $EM^2$ correlate with $SE$: negative $EM^2$ changes correlate positively while positive $EM^2$ changes 375 correlate negatively, i.e. both lead to reduced $SE$. The negative $EM^2$ changes correspond to "false alarms" in CTRL, which are corrected in the relaxation experiments. These "false alarms" potentially contribute to artificially higher $STN$ in CTRL despite low skill The changes in $ES^2$ on the other hand do not correlate with $SE$ (Table 4). Thus, while the ensemble spread, and so the uncertainty, is reduced in these areas in the relaxation experiment, and this reduction contributes to overall increased skill in the region, the magnitude of the error reduction in individual forecasts is not proportional to the magnitude 380 of the $ES^2$ decrease.

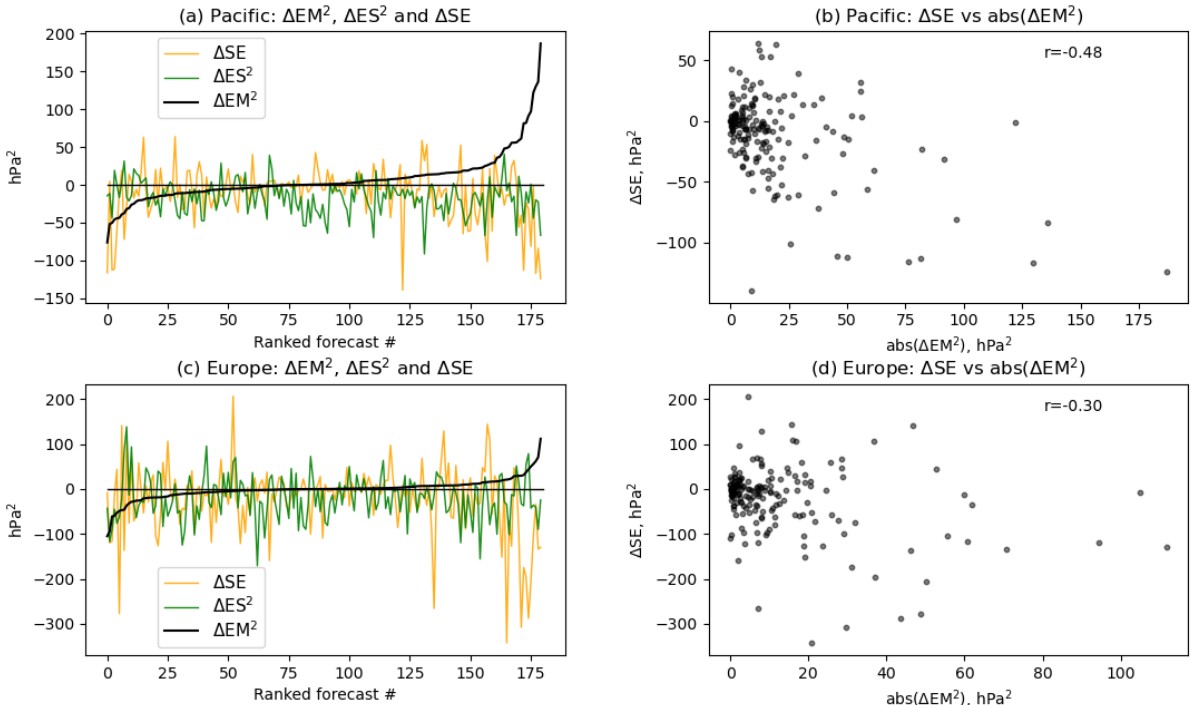

**Figure 7: Changes in EM$^2$, ES$^2$, and SE (hPa$^2$) in (a-b) Pacific (35°N-45°N and 155°W-125°W) in TROP and (c-d) Northern Europe (60°N-80°N and 20°E-60°E) in STRAT during weeks 5-6. (a,c) Changes in EM$^2$, ES$^2$, and SE in each individual forecast. The forecasts are ranked according to the change in EM$^2$ between the CTRL and the respective relaxation experiments: from the largest decrease to largest increase. (b,d) Scatterplots between absolute change in EM$^2$ and change in SE (hPa$^2$).**

The regions discussed above are selected to illustrate the two contrasting situations, but the results are not specific to the regions, nor to the type of the relaxation experiment. For example, the skill increase over southern Europe in STRAT coincides with both a decrease in $\sigma^2_{ES}$ and an increase in $\sigma^2_{EM}$, i.e. the situation there is similar to the example from TROP shown in Fig. 7(a-b).

Based on the analysis of Fig. 7 and Table 4 we highlight several important results regarding the northern Europe location: (i) variability of $EM^2$ does not increase between CTRL and STRAT but the skill does; (ii) a reduction of $EM^2$ in many individual forecasts leads to increased skill, which cannot be reconciled with the assumption that $EM^2$ is proportional to the magnitude of signal; (iii) $\rho^{STN}_{perf}$ (which is directly proportional to $\sigma^2_{EM}$) is considerably larger than $\rho$ in CTRL but they are closer to each other in STRAT. Taken together these results suggest that $\sigma^2_{EM}$ in CTRL overestimates the variability of the predictable signal, i.e. $EM$ does not represent the signal in CTRL sufficiently well. Note that, unlike $\rho^{STN}_{perf}$, $\rho^{ACC}_{perf}$ is

comparable to $\rho$ in CTRL but it strongly underestimate $\rho$ in STRAT demonstrating some sensitivity of the results to the definition of $\rho_{perf}$.

According to Eq. 11, as long as $M < \infty$, $EM$ is only an approximation for the predictable signal, and $\sigma^2_{EM}$ is contaminated with noise. The results for the northern Europe imply that the increased signal due to the stratospheric relaxation is balanced by reduced noise, and that the contribution of noise, the second term in Eq. 11, to $\sigma^2_{EM}$ in CTRL is comparable to the magnitude of the stratospheric signal. In contrast, in the Pacific location, the increased $\sigma^2_{EM}$ corresponds well with the increased skill (compare changes in $\rho$ and $\rho^{STN}_{perf}$ or $\rho^{ACC}_{perf}$ in Table 4), suggesting that it captures well the predictable signal, and the contribution of the noise is relatively small.

Both the level of noise and the size of the ensemble control how closely $\sigma^2_{EM}$ approximates the signal. While the level of noise cannot be changed, the size of the ensemble can be changed to ensure that $EM$ more reasonably captures the signal. We next estimate the size of the ensembles required to capture the tropical and stratospheric signals.

### 3.4 Ensemble size estimate for SLP

We use Eq. 13 to calculate the size of forecast ensemble required to ensure that $\sigma^2_{signal}$ represents at least 2/3 of $\sigma^2_{EM}$ in SLP, i.e. it is at least twice as large as the contribution of noise (the second term in Eq. 11). To estimate $\sigma^2_{signal}$ for the teleconnections ($\sigma^2_{telecon}$), we use Eq. 12 assuming that the teleconnection signal is given by the difference between the relaxation experiments and CTRL. This assumption is obviously not valid if CTRL captures the signals, which is the case especially for shorter lead times. To overcome this limitation, we focus on week 5-6 when the skill difference between the relaxation experiments and CTRL is largest, implying that the difference captures as large a fraction of the teleconnection signal as possible.

The results are shown in Fig. 8. Looking first at the results for TROP (Fig. 8a) one can see that capturing the tropical teleconnections across most of the sub-tropics requires an ensemble with ~10 members or even less. This is consistent with our previous analysis that showed a significant skill in CTRL and a strong increase of $\sigma^2_{EM}$ in TROP. The fact that there is a significant skill in CTRL indicates that Eq. 12 probably underestimates the strength of the teleconnection signal in the sub-tropics, and thus even smaller ensembles than what is suggested by our results are likely sufficient to capture the tropical teleconnections there. However, over most of Europe where the tropical signal is weak, >30 members, even ~100 members over northern Europe, are needed to capture the signal. For the stratospheric teleconnection (Fig. 8b), for most of the European and Atlantic region an ensemble size of 20-40 members is required for $\sigma^2_{EM}$ to be a reasonable representation of the stratospheric signal. The exceptions include the regions in northern Europe and the northern Atlantic close to 60°N, where the stratospheric SLP signal changes sign and is close to zero (e.g. Spaeth et al., 2024) and so large ensembles (~100 members), not necessarily available in most modern operational forecast systems, are required to extract the small signal. Note that the region over Northern Europe largely overlaps with the region analysed in Fig. 7(c-d), i.e. it represents an extreme case. The other exceptions are the two regions in subtropical Atlantic and around Greenland where ensembles with

10-15 members are sufficient. These regions probably correspond to the two nodes of NAO, and the relatively strong stratospheric influence there is well captured even by small ensembles comparable in size to the hindcasts used here. Scaife et al. (2014) showed that the seasonal NAO skill keeps increasing with ensemble size >20 members. Our results do not contradict this because our estimate only requires that the signal constitutes 2/3 of $\sigma_{EM}^2$. Further increase of the ensemble size would increase the contribution of the signal and so the skill.

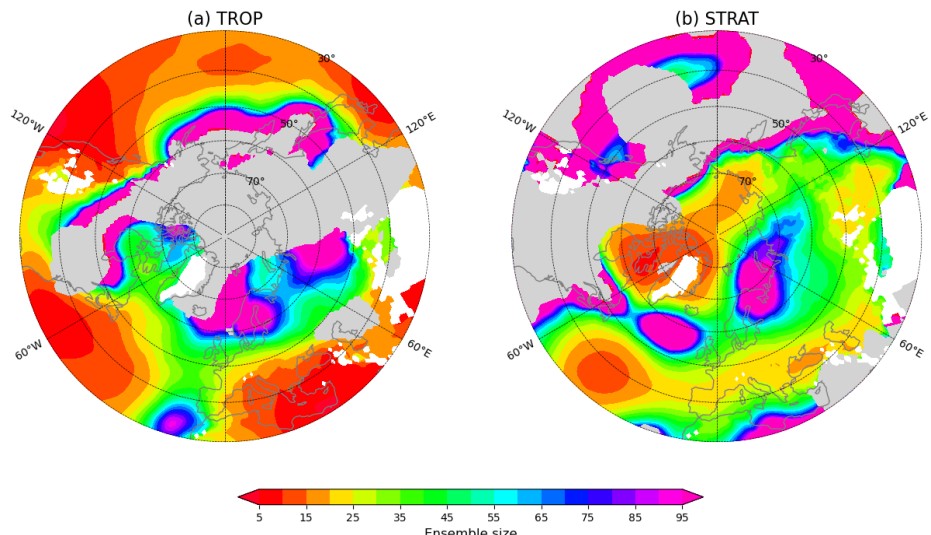

**Figure 8: Ensemble size required for at least 2/3 of $\sigma_{EM}^2$ in SLP to represent the (a) tropical and (b) stratospheric signal. Areas where the correlation skill increases by less than 0.05 between CTRL and the respective relaxation experiment are assumed to have no signal and therefore masked in grey.**

### 3.5 Results for surface temperature and total precipitation

We next look at the results for T2M and TP, the two parameters directly relevant for social and economic impacts. The T2M skill in CTRL (Fig. 9) is in general higher over the ocean than over the land and there are areas with high skill (>0.5) even during weeks 5-6. This is a consequence of the large thermal inertia of the ocean resulting in long persistence of the T2M anomalies. The area with high skill in the Barents-Kara seas lasting into weeks 5-6 is likely attributable to the variability of the sea ice edge. Similar regions can be seen in the Baffin Bay and northern Bering Sea during weeks 3-4. There are also areas over the land with high sub-seasonal skill. One possible explanation is that the skill is associated with persisting surface anomalies, e.g. from the snow cover. However, this is not consistent with the results of Richter et al. (2024) who found that, in their model, most of the skill at sub-seasonal timescales comes from the atmosphere, not from the land. Their result, however, can be model specific. The T2M skill distribution is spatially more heterogeneous than that of SLP, which reflects larger influence of the local factors such as orography and surface type.

While the skill generally increases in the relaxation experiments, the increases are small and mostly insignificant. In TROP, significant increases are mostly in the sub-tropics, but during weeks 5-6 they are also in the eastern US coast, eastern Europe and high latitudes around the date line. In STRAT, significant increases are found in eastern Europe, as well as in the Far East and north-eastern Canada. In these regions the temperatures are known to be affected by the NAO variability, so the skill is likely increased because of better represented NAO circulation. On the other hand, there is no skill increase across Northern Eurasia, the region known for a strong T2M response to SSWs (Sigmond et al., 2013; Butler et al., 2017; Domeisen et al., 2020) and strong polar vortex events (Tripathi et al., 2015).

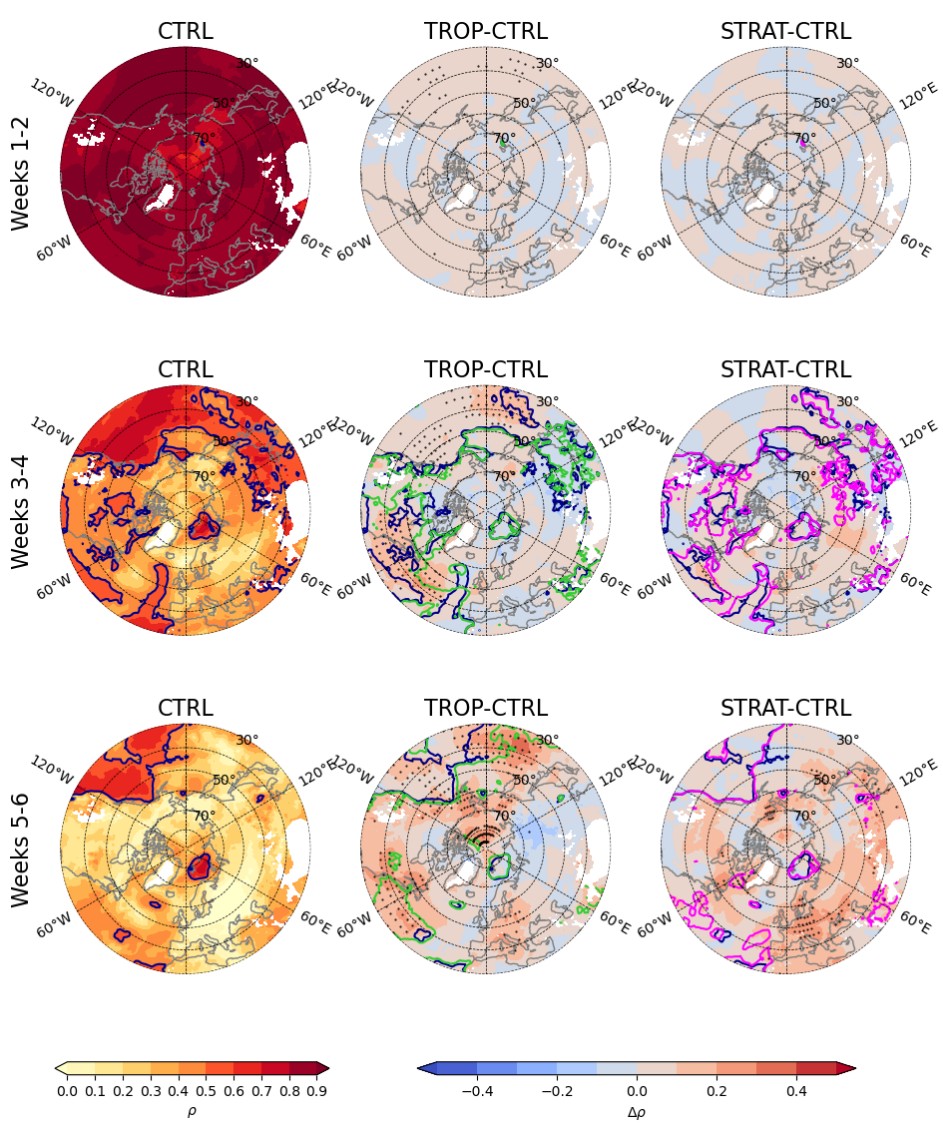

Figure 9: As Fig.1 but for T2M ρ skill.

The skill in TP drops much faster than that of SLP or T2M (Fig. 10). By weeks 3-4 the skill is small (<0.4) nearly everywhere. The increases in the relaxation experiments are small and strongly spatially heterogeneous. In TROP, significant increases at sub-seasonal time scales (Weeks 3-4 and 5-6) are seen in the sub-tropics over Pacific and Atlantic oceans, California, Japan, and also during Weeks 5-6 over the southwestern US and China. In STRAT, the skill is increased during Week 5-6 over the Atlantic coast of Spain, Morocco, southern UK, France and Norway, as well as over the Mediterranean. Like with the other variables, these increases are attributable to the increased NAO skill in STRAT (Fig. 2).

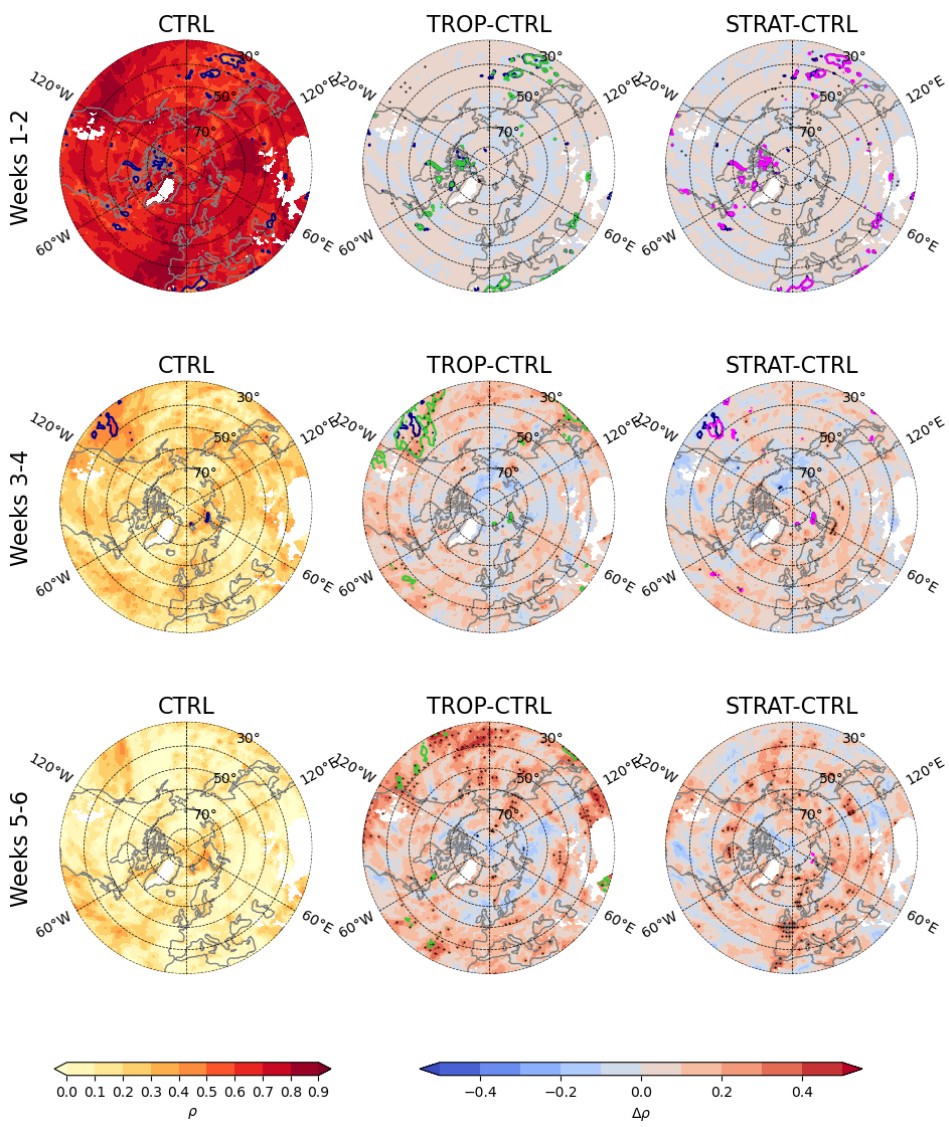

**Figure 10: As Fig.1 but for TP $\rho$ skill.**

Because the increases in T2M and TP skill in the relaxation experiments are rather small, the discussion of the changes in signal and noise is shortened, and the figures are given in supplementary materials (Figs. S2-S11). As for SLP, there is a correlation between $\rho$ and $\rho_{perf}$ (both $\rho_{perf}^{STN}$ and $\rho_{perf}^{ACC}$) at all lead times (Figs. S3-S4 and S8-S9), but for TP the correlation drops with lead times faster than that for SLP, and the relations break down for $\rho<0.4$. As for SLP, there is a weak correlation between changes in $\rho$ and $\rho_{perf}$ in TROP at weeks 3-4 and 5-6 for both T2M and TP (Tables S1 - S4), mostly because there are large areas in the sub-tropics where both $\rho$ and STN increase (Figs. S2 and S7), but the correlation is absent in STRAT. As for SLP, $\sigma_{EM}^2$ mostly increases (Figs. S5 and S10) while $\sigma_{ES}^2$ mostly decreases (Figs. S6 and S11) in the relaxation experiments, but unlike for SLP, these changes do not correlate with $\Delta\rho$, probably because $\Delta\rho$ for T2M and TP are smaller than those for SLP, and so more uncertain.

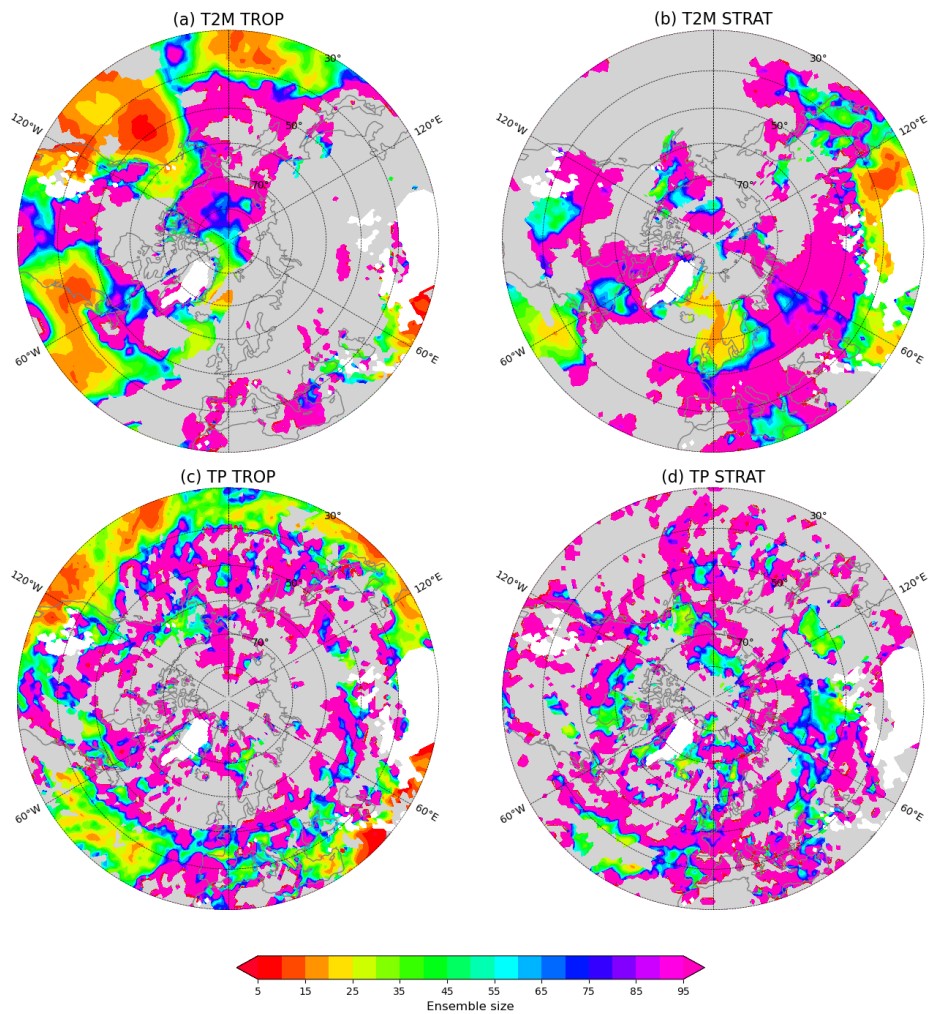

**Figure 11: Ensemble size required for at least 2/3 of $\sigma^2_{EM}$ in (a,b) T2M and (c,d) TP to represent the (a,c) tropical and (b,d) stratospheric signal. Areas where the correlation skill increases by less than 0.05 between CTRL and the respective relaxation experiment are assumed to have no signal and therefore masked in grey.**

Finally, we estimate the size of the ensembles required to capture the teleconnection signals in T2M and TP following the methodology outlined above for SLP (Fig. 11). The strong tropical signal in the Pacific requires 10-20 members to be well represented by $EM$, but the areas with such a strong signal are limited. Outside of the subtropics, the teleconnection signals are mostly weak and require large ensembles to be detected. One of the exceptions is the relatively strong STRAT signal in

western Europe which requires about 20 members to be detected in T2M over UK and Scandinavia, and 30-50 members to be detected in TP over Norway, UK and Spain. This signal is assumed to be associated with NAO variability. The stratospheric signal can also be detected with relatively few members in T2M over the eastern Mediterranean (40-50 members), China (10-30 members), and Japan (40-50 members), and in TP over southern Siberia (30-40 members). This latter TP signal has received relatively little attention in literature, although it appears to be a robust response to SSW (Butler

et al. 2017).

## 4 Discussion and conclusions

Atmospheric teleconnections can increase the forecast skill in the extratropical troposphere at sub-seasonal timescales by communicating the predictable signal from the stratosphere or the tropical atmosphere, the regions with enhanced predictability. Relaxation experiments used in our study demonstrate where and by how much the forecast skill could be

increased in winter season due to the teleconnections if the stratospheric and tropical variability were perfectly represented in the model. When looking at SLP, the tropical teleconnections mostly increase the skill in the subtropics and in the Pacific - North American region, in particular, the regions associated with the PNA. The stratospheric teleconnections affect the high latitudes and the regions affected by NAO variability. The predictability of the NAO found in the stratospheric relaxation experiment agrees well with a theoretical limit based on a simple signal-noise model (Charlton-Perez et al., 2021). Our study

focuses on winter season and the results do not necessarily apply to the other seasons because the influence of the teleconnections varies between seasons (e.g. Breeden et al., 2022).

According to our results, the teleconnections have smaller contributions to the skill of surface temperature and total precipitation compared to that of SLP, except in the subtropical oceans where the signal from the tropical teleconnections is strong. Both surface temperature and precipitation parameters are strongly affected by local processes (type of surface,

orography); therefore, even perfect representation of the teleconnections in the models would have only limited impact on their skill. Over Eurasia, stratospheric variability is known to affect both surface temperature and precipitation and the forecast models capture the link (Domeisen et al., 2020); however, demonstrating its influence on the skill has so far been difficult (Domeisen et al., 2020), consistent with our results. Nevertheless, small but significant increases in temperature and

precipitation skill are demonstrated over regions affected by NAO, mostly over Europe but also over Far East and northeastern North America.

Care must be taken in interpreting these results. Firstly, the difference between the relaxation and control experiments does not fully quantify the strength of the teleconnections because the teleconnections are represented to a certain degree already in the control forecasts. Secondly, the free running model will unlikely be able to achieve the level of skill present in the relaxation experiments. The relaxation experiments assume a perfect deterministic predictability in the nudging area, and therefore the impact of the nudged regions (the Tropics or the stratosphere) on the forecast skill is overestimated. On the other hand, if some processes are missing in the model to produce realistic teleconnections, then the impact on forecast skill will be underestimated.

An equally interesting question is how the skill increase in the relaxation experiments is related to changes in the forecast ensemble properties. Assuming, that the ensemble mean represents a predictable signal and the ensemble spread represents noise, an increase in the variability of the ensemble mean and a decrease in the ensemble spread are expected, and this is what we find over regions with a relatively large signal-to-noise ratio, such as the maritime subtropics. However, across most of the mid- and high-latitudes, where the signal-to-noise ratio at sub-seasonal timescales is low, the variability of the ensemble mean does not increase consistently with increased skill, suggesting that the ensemble mean is a bad approximation for a predictable signal there. The degree to which the ensemble mean approximates the signal depends on both the signal-to-noise ratio, and the size of the forecast ensemble. We propose a simple method to estimate the size of forecast ensembles required to capture the teleconnection signal, given the signal-to-noise ratio. More specifically, the method estimates the ensemble size required for the ensemble mean to capture a certain fraction of the signal. The method relies on estimating the teleconnection signal from the difference between the relaxation and control simulations, which demonstrates another application of the relaxation experiments. In subtropical regions with high signal-to-noise ratio, as little as ~10 members (or less) are sufficient for the ensemble mean to represent at least 2/3 of the tropical teleconnection signal in SLP. This size is comparable to the one used in our study, as well as in most other predictability studies that rely on the use of historical reforecasts. However, the size of ensembles required to capture weak teleconnection signals in most of the mid- and high latitudes (20-50 members) exceeds those typically available in the reforecasts. These ensemble size estimates are in reasonable agreement with those recently reported for seasonal forecasts based on analysis of the large ensembles (Koster et al., 2025). While the dependence of the skill on the size of ensemble is not a novel result, especially for long-range forecasts (e.g. Scaife et al., 2014; Butler et al., 2016), we believe that our estimates will raise the awareness about the potential limitation of small-sized reforecast ensembles which underestimate the skill of sub-seasonal forecasts. However, it should be emphasized that our method underestimates the strength of the teleconnection signal, and thus overestimates the required size of the ensembles, because it relies on the difference between the relaxation and control simulations while a part of the signal is captured by both simulations. While this effect is likely very small in the mid- and high latitudes where the control has virtually no skill during weeks 5-6, in the subtropics, where the control has skill, the required ensemble sizes are likely smaller than what our estimates suggest.

The result that small ensembles may not represent well the predictable signal associated with teleconnections implies, in particular, that such ensembles are not well suited for studies of windows of forecast opportunities. For example, Domeisen

et al. (2020) found that, although forecast models capture the signal associated with stratosphere-troposphere coupling, it does not lead to increased skill in T2M over Europe following weak polar vortex conditions often associated with enhanced predictability. In light of our results, we suggest that demonstrating increased skill might require larger ensembles than those available to Domeisen et al. (2020).

The comparison between control and nudged simulations allows us to detect issues that might appear because of small

ensemble sizes, but it does not mean that increasing ensemble sizes would eliminate other issues. Improving representation of the teleconnection (Erner and Karpechko, 2024; Afargan-Gerstman et al., 2024; Stan et al., 2022; Vitart and Balmaseda, 2024) is also important for improving sub-seasonal forecasting.

We have also investigated the relationship between the skill and ensemble spread. Recently, Spaeth et al. (2024) found a decrease in the ensemble spread following weak polar vortex events over Northern Europe suggesting reduced uncertainty of

the forecasts. In agreement with them, we also find a decrease in the spread in the stratospheric relaxation experiment, confirming a strong stratospheric influence in the region. The decrease in the spread apparently contributes to the skill increase in this region; however, we find no relationship between the magnitude of the spread decrease and reduction in forecast error across individual forecast. Thus, while a decrease in the spread can be an indicator of increased skill following some events, in general we find that the spread is not a reliable predictor of skill, consistent with previous studies (Barker,

1991). Again, hindcasts with ~10 ensemble members may be not well suited to address this question.

There is an ongoing discussion about whether the signal and noise at subseasonal and seasonal lead times are well represented in the forecast models with some studies highlighting the possibility that the level of noise in the models may be larger than it is in nature (Eady et al., 2014; Scaife and Smith, 2018; Garfinkel et al., 2024; Weisheimer et al., 2024). Although our results based on estimations of changes in signal and noise in nudged simulations do not exclude this

possibility, they do suggest that caution is needed when interpreting the estimates of sub-seasonal predictability in the extratropical troposphere obtained with small hindcast ensembles. We propose that the results based on small ensembles might need to be verified in the future when larger ensemble sizes that allow better separation of the signal and noise in the models will become available. Since 2023, operational ECMWF sub-seasonal ensemble forecasts have 101 members, and these may serve as a testbed for studies focusing on signal, noise and their relation to skill.


*Code and data availability.* Data from the nudging experiments are available from the ECMWF website under a Creative Commons Attribution 4.0 International license (CC BY 4.0). To view a copy of this license, visit https://creativecommons.org/licenses/by/4.0/. ERA5 reanalysis data can be downloaded from https://doi.org/10.24381/cds.bd0915c6 (Hersbach et al., 2020). A code repository to reproduce the data and the figures is in

preparation and will be made accessible upon publication.

*Author contribution.* AYK and AHB conceptualized the study, analysed the data and prepared figures; FV designed and carried out the nudging experiments; AYK wrote the original manuscript; AHB and FV contributed to the interpretation and discussion of the results and reviewed and edited the manuscript.


*Competing interests.* At least one of the (co-)authors is a member of the editorial board of Weather and Climate Dynamics.

*Acknowledgement.* AYK was supported by Research Council of Finland via grant no. 355792. We thank two anonymous reviewers for their valuable comments on earlier versions of the manuscript.

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
