# Peer review of "Signal, noise and skill in sub-seasonal forecasts: the role of teleconnections"

_EGUsphere, 2025_

## Author Comment (AC1)

Reply to Reviewers comments on the manuscript "Signal, noise and skill in sub-seasonal forecasts: the role of Teleconnections" by A. Karpechko et al.

We thank the anonymous Reviewers for their helpful comments. Below we provide point-to-point responses indicating how the manuscript has been revised. The comments by the Reviewers are repeated in *blue italic*, citations from the revised manuscript are in *black italic*:

**Reviewer 1**

*This study uses a set of ensemble relaxation experiments to explore the relationship between tropical and stratospheric teleconnections, forecast skill, and signal to noise relationships. Relaxing either the tropics or the stratosphere increases the forecast skill for SLP, and to a lesser degree for T2m and precip, in many regions; these effects are mostly consistent with previous work. The novel part is that the study then tries to diagnose whether the increases is associated with a signal in the ensemble mean, with a reduction in the ensemble spread, or both. While in many regions the answer is "both", there are numerous exceptions (including the Northern Europe signal in SLP to stratospheric nudging, where the ensemble mean signal is weak, and most of the skill increase comes from a reduction in ensemble spread). The authors then diagnose how big an ensemble is needed before it possible to reliably extract signal from noise, and find that larger ensembles than are used in this study would be needed to identify sub-seasonal predictability; this last part is where I think the study could be improved the most. Overall, the required revisions could be relatively minor if the authors decide to tone down the statements I found most objectionable, or more major if they disagree with my assessment and provide additional evidence supporting their statements. Either way, revisions are needed before I consider the final version.*
*There are three major comments that are somewhat related to one-another and concern how to interpret the signal to noise metrics presented in this paper:*
*1a. As alluded to above, I think the conclusions drawn from the analysis on the minimal ensemble size are likely overstated. I am particularly bothered by lines 23-24 in the abstract and 71-73 in the introduction. The discussion section (lines 513-518) is a little more careful, but even there I think the wording can be refined.*
*The minimal ensemble size used in this paper is true for the S2N definition and perfect model definition used here. But there are other ways of extracting subseasonal signals from forecast ensembles and skill can be demonstrated from much smaller ensembles in many situations.*
*Using long hindcasts we can extract teleconnection signals from the tropics using <5 ensemble members (e.g. Stan et al 2022). Domeisen et al 2020 (already cited) also showed that <5 members is enough to extract signals from the stratosphere for many models. Both of these studies use long hindcasts from several models, and demonstrate some skill at representing teleconnections using far fewer members, even as the skill will of course increase as ensemble sizes increase. I think the authors' results are demonstrating that signal exceeds noise only for ensemble sizes larger than 20, and such a signal to noise analysis is essential for deciding on ensemble size of real-time operational forecasts. But real-time forecasts use 50 members or more at least for IFS, so it would seem that operational forecasts are already large enough to extract signals in most regions. It would seem that rewording the text in the three locations noted above would be enough to resolve this issue, unless the authors disagree with me in which case additional work is needed.*

In general we agree with the point made by the reviewer. We believe the issue is in terminology, and perhaps there is some confusion about the terms "signal" and "extract signal". We would like to emphasize that the key point is that the predicted mean value of a finite forecast

ensemble contains both signal and noise, and this is hardly arguable. The larger the ensemble size, the larger the signal-to-noise ratio. Both studies mentioned by the Reviewer (Domeisen et al. 2020 and Stan et al. 2022), as well as many others, do show skill of the subseasonal forecasts associated with teleconnections, implying that even small ensembles can be enough to "extract teleconnection signals". This, however, does not mean that all predictable signals have been extracted in the above examples, and that a larger signal-to-noise ratio is not achievable. Our point is that the size of the ensembles is an important factor limiting forecast skill, and we do not think this contradicts the results of the above studies. However, reflecting on our original text, we see that the expression "extract signal" requires careful definition to avoid misinterpretation. In the revision we focus on avoiding possible misinterpretations. Specifically, we rewrite the three statements referred to by the Reviewer as follows:

Lines 20-24: "*We suggest that the ensemble size available in the experiments (11 members) is not always enough to make it possible to fully extract signal from noise, and that larger ensembles (20-50 members or even more depending on area and variable) would be beneficial for studies of sub-seasonal predictability associated with the teleconnections in mid- and high latitudes, including windows for forecast opportunities.*"

Lines 71-73: "*We will further argue that in many cases extracting teleconnection signals in mid- and high latitudes may require larger ensemble sizes than are in most datasets available for research (Vitart et al., 2017); and provide some estimations of what ensemble sizes are required for capturing a substantial fraction of the signals associated with the stratospheric and tropical teleconnections.*"

Lines 508-518: "*We propose a simple method to estimate the size of forecast ensembles required to capture the teleconnection signal, given the signal-to-noise ratio. More specifically, the method estimates the ensemble size required for the ensemble mean to capture a certain fraction of the signal. The method relies on estimating the teleconnection signal from the difference between the relaxation and control simulations, which demonstrates another application of the relaxation experiments. In subtropical regions with high signal-to-noise ratio, as little as ~10 members (or less) are sufficient for the ensemble mean to represent at least 2/3 of the signal in SLP. This ensemble size is comparable to the one used in our study, as well as in most other predictability studies that rely on the use of historical reforecasts. However, the size of ensembles required, according to our estimates, to capture weak teleconnection signals in most of the extratropics (20-50 members) exceeds those available in the reforecasts. These estimates are in reasonable agreement with those recently reported for seasonal forecasts based on analysis of the large ensembles (Koster et al., 2025). While the dependence of the skill on the size of ensemble is not a novel result, especially for long-range forecasts (e.g. Scaife et al., 2014; Butler et al., 2016), we believe that our estimates will raise the awareness about the potential limitation of small-sized reforecast ensembles which might underestimate the skill of sub-seasonal forecasts.*"

In addition, we have toned down some other statements in the manuscript, to make sure the language reflects our findings correctly, and to avoid misinterpretation.

*1b. A related issue is that equations 12 and 13 work in the limit that Control has no skill. If I understand equation 12 and 13 correctly, the residual skill in CTRL in week 5-6 will lead to an overestimate of the minimal ensemble size. This is because of nonzero sigma^2 in CTRL. Is there a way to account for this effect in the derivation of equation 13, or at least quantify how important this effect might be?*

The reviewer is right that Equations 12 and 13 work in the limit that the control has no skill, and that the residual skill in the control means that the magnitude of the teleconnection is larger than what is given by Eq. 12. Consequently, the ensemble size required to capture this signal is smaller than what is given by Eq. 13. This has been discussed in the original manuscript in lines 393-398.

We do not know how to account for this effect explicitly because Eqs. 12-13 do not include skill. Moreover, there might be other sources of skill in CTRL, not related to teleconnections (e.g. persistence of SST anomalies). We believe that in the extratropics where the correlation skill in the control during weeks 5-6 is below 0.2 (and often below 0.1), the effect is small, or even negligible. The results are more sensitive to the residual signal of the control in the subtropics, where the correlation skill of the control is about ~0.3, and thus, the control likely captures a sizable fraction of the teleconnection signals. These are the same regions where, according to our estimates, the required ensemble size is relatively small. To be more explicit about the shortcoming of our method we add the following comments to the text:

In Section 2.3: "*Note that if a part of the teleconnection signal is captured by CTRL, then Eq. 12 would underestimate the magnitude of the signal.*"

In Section 3.4: "*The fact that there is a significant skill in CTRL indicates that Eq. 12 probably underestimates the strength of the teleconnection signal in the sub-tropics, and thus even smaller ensembles than what is suggested by our results are likely sufficient to capture the tropical teleconnections there.*"

In the Discussion: "*However, it should be emphasized that our method underestimates the strength of the teleconnection signal, and thus overestimates the required size of the ensembles, because it relies on the difference between the relaxation and control simulations while a part of the signal is captured by both simulations. While this effect is likely very small in the mid- and high latitudes where the control has virtually no skill during weeks 5-6, in the subtropics, where the control has skill, the required ensemble sizes are likely smaller than what our estimates suggest.*"

*1c. An alternate way of thinking about "perfect model" and signal to noise is the ratio of predictable components (RPC) from Smith and Scaife 2018 (already cited). This definition seems to be more robust to ensemble size, and can identify S2N issues with relatively small ensembles (see figure 1 of Smith and Scaife and figure S17 of Garfinjkel et al 2024; already cited) though bigger ensemble sizes certainly help. I hate to add yet another metric to this already comprehensive paper, but I think the authors need to compute RPC if they really think their statements in the three locations outlined above are correct. Otherwise, the statements in the abstract and end of discussion about minimum ensemble size need to be made more specific to one specific method of ascertaining signal to noise. On a related note, it isn't clear to me whether RPC and S2N metrics are actually the same thing, or even closely related, despite the fact that they both use similar terminology; hence the closing paragraph on lines 535-540 seems overly speculative at the moment.*

Please note that we do not explicitly consider RPC in this paper, partly because we want to keep the paper concise, and partly because of the confusion regarding the RPC definition. In particular, the recent paper by Weisheimer et al. (2024, cited in the manuscript), states that "The RPC is highly sensitive to the relative magnitudes of the variances of the observations, the ensemble-mean forecast and the error of the ensemble mean. When coupled with weak correlations and short observed records, this results in high sampling uncertainties that can

significantly impact the RPC estimates and compromise their robustness." However, we have tested the alternative definition of the perfect model correlation skill given in Scaife and Smith 2018. Perfect model correlation is the denominator in RPC, and the way it is estimated strongly controls the value of RPC.

In particular, we have found that using this alternative definition of the perfect model correlation affects some conclusions emerging from Figure 4. Figure R1 repeats Figure 4 from the manuscript but with the perfect skill calculated following the definition used in Scaife and Smith 2018. Figure R1 confirms that there is a correlation between $\rho_{perf}$ and $\rho$ fields which remains significant until weeks 5-6 although it declines with time. It also confirms that the relationship between $\rho_{perf}$ and $\rho$ breaks for low $\rho$; however, in Figure R1 this effect becomes visible only for $\rho<0.3$. The biggest effect of the different definition of $\rho_{perf}$ is in the absolute values of $\rho_{perf}$, which are lower for this definition than for the definition used in the original manuscript. Importantly, the statement "the perfect skill is mostly larger than the actual skill (i.e., the values fall below the diagonal line)" is not valid when the new definition of the perfect skill is used. The values are more evenly distributed around the diagonal line. These new results will be incorporated into the revised manuscript. The results regarding the changes in $\rho_{perf}$ and $\rho$ in the relaxation experiments are not affected by the different definition of $\rho_{perf}$.

[Figure]

Figure R1: The same as Figure 4 of the manuscript except that the perfect correlation is calculated as in Scaife and Smith 2018.

*Otherwise, the statements in the abstract and end of discussion about minimum ensemble size need to be made more specific to one specific method of ascertaining signal to noise. On a related note, it isn't clear to me whether RPC and S2N metrics are actually the same thing, or even closely related, despite the fact that they both use similar terminology; hence the closing paragraph on lines 535-540 seems overly speculative at the moment.*

RPC and S2N metrics are both based on estimates of signal and noise, and thus they are related. Concerning the last paragraph in the discussion, we agree with the reviewer and have modified the manuscript as follows:

"*There is an ongoing discussion about whether the signal and noise at subseasonal and seasonal lead times are well represented in the forecast models with some studies highlighting the possibility that the level of noise in the models may be larger than it is in nature (Eady et al., 2014; Scaife and Smith, 2018; Garfinkel et al., 2024; Weisheimer et al., 2024). Although our results based on estimations of changes in signal and noise in nudged simulations do not exclude this possibility, they do suggest that caution is needed when interpreting the estimates of sub-seasonal predictability in the extratropical troposphere obtained with small hindcast ensembles. We propose that the results based on small ensembles might need to be verified in the future when larger ensemble sizes that allow better separation of the signal and noise in the models will become available. Since 2023, operational ECMWF sub-seasonal ensemble forecasts have 101 members, and these may serve as a testbed for studies focusing on signal, noise and their relation to skill.*"

*(Given the fact that STRAT nudging is increasing skill in Northern Europe despite not increasing EM variability, I strongly suspect there is an RPC>1 issue in this region. This is likely to be similar to the RPC>1 issue shown by Garfinkel et al 2024 for this model in polar cap height)*

This is an interesting suggestion. We have tried using the definition of the perfect skill used in Scaife and Smith 2018 for the Northern Europe example discussed in the text. For the control ensemble, the mean perfect skill (i.e. the average of individual members correlation skills) is 0.07 for weeks 5-6 with individual members correlation skills varying between -0.06 and 0.21. The correlation with ERA5 is 0.05. This means that the model predicts itself as good, or even slightly better, than it predicts ERA5. Thus, RPC is close to 1, which is the expected result. For the stratospheric nudging experiment the situation is different. The mean perfect skill is 0.13 with individual members skills varying between 0.0 and 0.26. The correlation with ERA5 is 0.42, well outside the range of the perfect model skills. This noticeable difference might have a simple explanation –we selected this area because of a large increase in skill, which is probably subject to sampling issues. Explaining these results in terms of statistical problems (small ensemble size, insufficient sampling) seems reasonable. Otherwise it is difficult to explain why the model does not have RPC anomalies in the control but does have it in the nudged simulation.

*Minor comments*
*Line 19/20: an additional possibility is that the model isn't fully utilizing the predictable signal, or possibly is misrepresenting the predictable signal.*
Yes, the model has errors/biases which do not allow it to fully utilize the predictable signal due to the teleconnections, or even to misrepresent it. However, the statement in question refers to the point that the EM variability in control and in STRAT are nearly the same, despite reduced noise and increased skill in STRAT. This implies that the EM variability in the control is larger than the variability of the predictable signal. Following this, and the other comment below

(L380-381) we added a statement to the discussion section, to acknowledge that it is also important to improve the representation of the teleconnections in the model:
"*The comparison between control and nudged simulations allows us to detect issues that might appear because of small ensemble sizes, but it does not mean that increasing ensemble sizes would eliminate other issues. Improving representation of the teleconnection (Erner and Karpechko, 2024; Afargan-Gerstman et al., 2024; Stan et al., 2022; Vitart and Balmaseda, 2024) is also important for improving sub-seasonal forecasting.*"

*Line 44: missing word in "some state-of-art can capture"*
Corrected: "some state-of-art models can capture"

*Line 58: I suggest adding Stan et al 2022*
Thank you for this suggestion. Indeed, this reference is useful and has been added.

*Table 1: is there tapering for the stratospheric nudging below*
Yes, we have added to the text "with tapering starting above 70 hPa"

*Line 139: the "(\rho)" belong two words earlier in the sentence*
We prefer to keep the "$(\rho)$" where it is, because the anomaly correlation coefficient can be calculated for time series as well as for spatial fields. In this case, the "$(\rho)$" refers specifically to the anomaly correlation coefficient calculated for time series.

*Line 380-381: is it possible to provide a more physically meaningful interpretation? For example, is there overly strong downward coupling from the stratosphere to Northern Europe in control?*
If by overly strong downward coupling the reviewer means some biases in the model then this would be equally an issue in all simulations, nudged and control, because the same model is used in both experiments. In our study we focus on the differences between the nudged and control simulations in particular because this eliminates the issues common to both simulations, such as possible biases in the stratosphere-troposphere coupling. Nudging constraints the model and reduces noise, thus allowing the model to capture the signal better than it does in the unconstrained control run. To acknowledge the importance of physical biases we add the following sentence in the Discussion section:
"*The comparison between control and nudged simulations allows us to detect issues that might appear because of small ensemble sizes, but it does not mean that increasing ensemble sizes would eliminate other issues. Improving representation of the teleconnection (Erner and Karpechko, 2024; Afargan-Gerstman et al., 2024; Stan et al., 2022; Vitart and Balmaseda, 2024) is also important for improving sub-seasonal forecasting.*"

*Figure 8 and similar other figures: suggest masking regions without skill with a different color than white, since white is used for topography.*
Done. Thanks for the suggestion!

**Reviewer 2**

*The paper uses a series of forecast model temperature-nudging experiments to investigate how do atmospheric teleconnections from the stratosphere and the tropics influence the forecast skill at subseasonal timescales. Specifically, the study examines whether the increase in the forecast skill in the relaxation experiments is reflected in the variation of the ensemble mean or its spread (or both), by separating between the (predictable) signal and the (unpredictable)*

*noise. Results show that in the stratospheric relaxation experiments, the increase in skill in high-latitude is not reflected in an increase in the ensemble mean. The authors conclude that extracting signal from noise requires a larger ensemble size than to the ensemble size used in this study.*

*Overall, the study performs a comprehensive analysis to extract signal from noise and understand subseasonal predictability skill and its sources. While the work is concise and well-written, I was not convinced about the reliability of the signal-noise model presented in this study. Therefore, a major revision is required in order to address several major issues (as described in detail below).*

***Main comments:***

*1. The study examines how does nudging tropical and stratospheric temperature and wind fields influence the forecast skill, ensemble variability and ensemble spread at subseasonal timescales. It raises the question whether using variances of the ensemble mean and ensemble spread are valid representations of the predictable (signal) and unpredictable (noise) variances of the model. However, as the authors themselves say "in general these are not the same things". I am not convinced yet that these definitions reliably represent what they intend. In particular, the signal (EM) is defined as deviation from hindcast climatology (eq. 3), thus its variance (and well as the ensemble spread variance) represents the model's variability, whereas the anomaly correlation coefficient (ACC) is defined with a 'reference' of ERA5 climatology. This ACC is later compared to STN, however – are they comparable? It would be good if the authors justify this approach and why do we expect STN to be directly comparable to ACC.*

If the model had no structural errors, the ensemble members generated by such a model would be statistically identical to the real-world realization, which in our study is represented by ERA5 reanalysis. Thus, the forecast ensemble members would share the same signal as the real-world realization (ERA5). In such a model, higher STN, estimated from the statistical properties of the forecast ensemble, would imply that individual ensemble members are more similar to each other, as well as to ERA5, because they share the same signal. Conversely, lower STN would indicate less similarity between the individual forecast members, and between forecast members and ERA5. Since ACC is a measure of similarity, in such a model we expect that a higher/lower STN corresponds to a higher/lower ACC. But even if a model has structural errors (as they all do), one still can expect a certain degree of correspondence between STN and ACC, as long as the model has skill, because the model and the real world share the same signal. Thus, testing a correspondence between STN and ACC looks logical to us. Following the Reviewer's advice, we make this point clearer in the text:

"*While $\sigma_{EM}^2$ is the best estimate for $\sigma_{signal}^2$ and $\sigma_{ES}^2$ is the best estimate for $\sigma_{noise}^2$, in general these are not the same things because the models have structural errors and because the ensembles have a finite size. Yet, as long as the model has skill in predicting the real world, a correspondence between signal-noise ratio estimated from the properties of the forecast ensemble and the forecast skill is expected. Consequently, one can ask whether a forecasted large signal-noise ratio (either due to an anomalously large EM or anomalously small ES) indicates an enhanced predictability and skill.*"

*2. Using the same logic, a high false alarm rate, for example, may lead to an increased signal to noise ratio, since the STN is based on the hindcast climatologies, but it is not an indication*

*of a good skill. Therefore, STN would be sensitive to false alarms in the model, and this can suggest another expiation for high skill in regions with low ACC.*

While "false alarm rate" may not be the proper word for the forecasts of continuous variables, we see what the Reviewer means, and we fully agree with this point. In fact, we believe that in the manuscript we describe the same effect as the Reviewer. To make it clear we added the following statement to the manuscript:

"*The negative $EM^2$ changes essentially correspond to "false alarms" in CTRL, potentially contributing to artificially higher STN despite low skill, which are corrected in the relaxation experiments.*"

*3. In this study, 'signal' is defined based on anomalies from the hindcast climatology. This definition means that large variance of SLP anomalies will contribute to increased 'signal to noise' ratio (and possibly an improved skill), while smaller variance may not. However, if a model predicts conditions that are similar to climatology – with variance comparable to the climatological variance – what would STN represent in that case?*

We assume that similarity to climatology is measured in terms of variance units, rather than in absolute units. It is a common approach to define "signal" using the variance of the ensemble mean (e.g. Eade et al. 2014) and we follow this approach in our study. If the ensemble mean predicts conditions similar to climatology, then the "signal" is small. If, at the same time, the total variance in the region is large, then STN is low. Note that STN and ACC are estimated using time series of forecasts. Discussing individual forecasts in terms of signal and noise should be made with caution because skillful forecasts can be made in a region with low STN, and vice versa. We believe this comment does not require changes in the text.

*Another related question: what does the high STN in the CTRL experiment in week 1-2 represent (Figure 3), e.g. is it close to 1 because of the small ensemble spread or due to a large variance of EM? Or both?*

Yes, it is both the small ensemble spread and the large variance of EM that contribute to high STN during weeks 1-2. This can be clearly seen by comparing Figures 5 and 6 which have the same units.

*4. Is there a possibility that the ensemble agreement (reduced ES variance) may capture only a certain type of skill improvements, e.g., following SSW events. Could it be that the STN model can be a good reflection of the skill for such episodes rather than for all the initializations? Analyzing specific episodes could be a useful way to test if teleconnection-based skill really occurs simultaneously with STN increase, e.g., for a case study. This may help to clarify concerns and justify the STN model approach for skill.*

Yes, a decreased spread can probably indicate increased skill in some cases, for example following weak vortex events as reported by Spaeth et al. (2024). In fact, we also mention in the introduction the study by Vitart et al. (2025) which finds that some state-of-art models can capture the spread-skill relationship at sub-seasonal timescales. However, we did not find this relationship in our study. In particular, we looked at subsets of weak polar vortex cases and while we find a reduction in the spread following these events on average, we do not see a correspondence between spread and skill across individual events. The lack of spread-skill correspondence possibly can be because of too small ensemble size. We modify the statement in the conclusions to reflect the comment by the Reviewer as follows:

*"Thus, while a decrease in the spread can be an indicator of increased skill following some events, in general we find that the spread is not a reliable predictor of skill, consistent with previous studies (Barker, 1991). Again, hindcasts with ~10 ensemble members may be not well suited to address this question."*

*5. Fig.4 suggest that the actual skill is a function of the "perfect model" skill only in the CTRL run and at short lead times. In the relaxed simulations – this relation does not seem to be as linear as for the CTRL. However, the interpretation of this result is that the STN ration does not reflect the improved skill in the relaxation experiments. However, the authors do not provide an alternative explanation for this outcome. This may also relate to the previous questions – does the STN model, as defined here, accurately represent skill changes?*

We would like to clarify that the actual skill and the perfect skill behave in the relaxation experiments similarly to that in CTRL. A statement to this point is added to the revised manuscript as follows:

*"The relationship between the $\rho$ and $\rho_{perf}$ fields in TROP and in STRAT behave similarly to that in CTRL (not shown)."*

The discussion that follows in the text concerns the relationship between the *changes* in the perfect skill ($\Delta\rho_{perf}$) and the *changes* in the actual skill ($\Delta\rho$), and, as the Reviewer comments, we note that the changes in the perfect skill (or in STN) do not follow those in the actual skill. We believe our manuscript does provide an explanation to this point. In particular, we state in the Abstract that *"In high latitudes, where the stratospheric impacts are strongest, the EM variability does not increase in the stratospheric relaxation experiments consistently with increases in skill, implying that EM does not reflect the predictable signal."* This implies that STN estimated from the forecast ensemble does not particularly well reflect STN in the real world. We further present estimates, according to which the ensemble size used in our study may be too small to fully extract the small signal associated with the teleconnections.

Similarly to the Reviewer we wonder about how well the STN model reflects the changes in the actual skill, and as we replied to Comment #1, we believe it is a logical question which we address in this study. Below we repeat the part of the text which we modified following Comment #1, and we hope these changes are sufficient to address the concern of the reviewer:

*"While $\sigma_{EM}^2$ is the best estimate for $\sigma_{signal}^2$ and $\sigma_{ES}^2$ is the best estimate for $\sigma_{noise}^2$, in general these are not the same things because the models have structural errors and because the ensembles have a finite size. Yet, as long as the model has skill in predicting the real world, a correspondence between signal-noise ratio estimated from the properties of the forecast ensemble and the forecast skill is expected. Consequently, one can ask whether a forecasted large signal-noise ratio (either due to an anomalously large EM or anomalously small ES) indicates an enhanced predictability and skill."*

*6. Fig.3 shows that STN is largest in subtropical ocean basins – the Pacific and western Atlantic. This is not surprising as these are the storm track regions – where the variance of many atmospheric variables reaches their peak (on daily and weekly timescales; see Fig.2 in Chang et al., 2002). I think this goes back to the definition of STN, and the question whether it is right to represent the "signal" as the variance of SLP anomalies from hindcast climatologies (eq. 3). Please make sure that the definition of STN is not simply an indication of the regions with largest variance in winter.*

Thank you for this comment. Yes, we make sure STN is not an indication of the region with largest variance in winter. Figure 2c by Chang et al (2003) clearly shows that the storm tracks, as defined using SLP variance, coincide with the areas where both $\sigma^2_{EM}$ and $\sigma^2_{ES}$ calculated in our study maximize (Figures 5 and 6 of our manuscript respectively). We interpret these $\sigma^2_{EM}$ and $\sigma^2_{ES}$ maximums as indicators of the storm track locations in winter, and we refer to Chang et al. (2003) in the revised manuscript to support this interpretation. Note that these areas are broadly located around 50°N (40°N-60°N).

However, STN maximums are located further south from these storm tracks by some 10°, mostly between 30°N and 50°N, as can be seen in Figure 3. The spatial correlation coefficients between STN and $\sigma^2_{EM}$ or $\sigma^2_{ES}$ fields (Table 2 of the original manuscript) are small, further supporting our statement that STN does not coincide with the areas of maximum SLP variability (marked by $\sigma^2_{EM}$ and $\sigma^2_{ES}$ maximums). While there is partial overlap between STN and $\sigma^2_{EM}$ fields (correlation coefficient ~0.3) the correlation between STN and $\sigma^2_{ES}$ is lacking. At the same time the correlation between $\sigma^2_{EM}$ and $\sigma^2_{ES}$ fields is large (~0.8) as they both reflect the areas of maximum SLP variability. Since STN is a ratio of $\sigma^2_{EM}$ and $\sigma^2_{ES}$ it is not a surprise that the maximum of STN does not coincide with the maximums in either $\sigma^2_{EM}$ or $\sigma^2_{ES}$. In summary, there is no evidence that STN reflects the location of the storm tracks. We add the reference to the storm tracks as follows:

"*The three regions where $\sigma^2_{EM}$ maximizes are the climatological Icelandic and Aleutian lows coincident with the location of the Northern Hemispheric storm tracks (Chang et al., 2002), and the Ural high, and these are the same regions where $\sigma^2_{ES}$ maximizes too. Overall $\sigma^2_{EM}$ and $\sigma^2_{ES}$ fields correlate strongly positively with each other at all lead times but neither of them correlates with $\rho_{perf}$ or STN, with correlation coefficients ranging between -0.3 – 0.3 (Table 2).*"

**Minor/technical comments:**

*Line 122: The variability -> the variance?*

Yes, corrected.

*Line 245: the authors mention that the mean downward coupling is usually small over the Pacific region (Dai et al., 2023), and therefore it is not clear how relaxation of the stratosphere contributes to the increased PNA skill. First, Dai et al has analyzed sudden stratospheric warming events, and therefore, focused only on specific episodes. Second, is it possible that STRAT overestimates the Pacific response to stratospheric variability, and the PNA response in particular. This can be easily examined by comparing to the free running CTRL.*

We agree it is important to specify that Dai et al analyzed SSW events and we have modified the relevant statement accordingly. Concerning the second part of the comment, please note that in this paragraph we discuss enhanced PNA skill in the relaxation experiment. If the model overestimated the Pacific response to stratospheric variability in STRAT, this would lead to decreased skill, not to increased skill. Thus, we do not think that it is possible that the model overestimated the response. We respond to this comment by modifying the text as follows:

"*Since the zonal mean downward coupling following Sudden Stratospheric Warmings has usually small impacts over the Pacific region (Dai et al., 2023) it is not clear which processes contribute to the increased PNA skill.*"

*Line 260: "actual correlation skill" -> perhaps rephrase the term "actual", e.g., model's skill*

The term "actual" has some history of use (see e.g. Kumar 2014) and we prefer to keep it to be consistent with the literature. We change the term "actual correlation skill" to "model's actual skill" and hope this change is sufficient to address reviewer's concern.

*Line 235: using -> used*

We believe that "using" is grammatically correct. In the revised version we separated the participial phrase with commas, which hopefully makes the sentence clearer:

"*Charlton-Perez (2021), using a statistical model, estimated that..*"

*Line 500: "A not less interesting question" – rephrase.*

We changed it to: "*An equally interesting question…*"

*Line 500: "these skill increases" –> this skill*

We rewrote the sentence as follows:

"*An equally interesting question is how the skill increase in the relaxation experiments is related to changes in the forecast ensemble properties.*"

*Line 540 "We propose that some conclusions regarding the predictability of the extratropical troposphere … might need to be revised in the future when larger ensemble sizes required to correctly separate the signal and noise in the models will become available" – this statement is too general. Ca the authors specify what are "some conclusions"?*

Following the suggestion by the other reviewer, this section has been rewritten. The revised version does not contain this too general statement. We repeat it here for convenience:

"*There is an ongoing discussion about whether the signal and noise at subseasonal and seasonal lead times are well represented in the forecast models with some studies highlighting the possibility that the level of noise in the models may be larger than it is in nature (Eady et al., 2014; Scaife and Smith, 2018; Garfinkel et al., 2024; Weisheimer et al., 2024). Although our results based on estimations of changes in signal and noise in nudged simulations do not exclude this possibility, they do suggest that caution is needed when interpreting the estimates of sub-seasonal predictability in the extratropical troposphere obtained with small hindcast ensembles. We propose that the results based on small ensembles might need to be verified in the future when larger ensemble sizes that allow better separation of the signal and noise in the models will become available.*"

---

## Author Response (AR2)

Reply to Reviewer comments on the manuscript "Signal, noise and skill in sub-seasonal forecasts: the role of Teleconnections" by A. Karpechko et al.

We thank the anonymous Reviewer for helpful comments. Below we provide point-to-point responses indicating how the manuscript has been revised. The comments by the Reviewer are repeated in *blue italic*, citations from the revised manuscript are in *black italic*:

**Reviewer 2**

*The main aim of the manuscript is to investigate how the atmospheric teleconnections from the stratosphere and the tropics affect the signal and noise (and their ratio) in subseasonal forecasts. Results show that teleconnections affect the signal-noise ratio if there is a sufficiently large signal-noise ratio. However, in mid- and high-latitudes, the signal-noise ratio becomes too small on seasonal timescales. The authors conclude that extracting the effect of teleconnection requires larger ensemble sizes than are in most datasets available for research and provide an estimation of the required ensemble size for capturing the signal (or response) associated with the stratospheric and tropical teleconnections.*
*The authors have addressed my main concerns in the revised version of the manuscript. I recommend this paper for publication.*

Thank you!

*(Specific technical points are listed below)*
*Acronym ACC is not defined in the text.*
ACC is defined in the revised manuscript.

*Abstract: "Skill improvements are considerably smaller for surface temperature and total precipitation, suggesting a smaller role of the teleconnections in their predictability"-However, is it considerably smaller or comparable? (see Figs 9-10, and even Fig.11). It seems that there is an improved skill for temperature and precipitation as well (although smaller), although scattered.*
SLP skill is increased by more than 0.4 over sizeable areas in both experiments, and even by 0.5 in TROP during weeks 5-6. Such large increases are nearly missing in T2M and TP fields, so we still believe that the improvements are smaller for T2M and TP. Given the reviewer's concern, we remove word "considerably", so that the revised text reads: "*Skill improvements are smaller for surface temperature and total precipitation...*"

*Line 400: Eastern Pacific and northern Europe - include coordinates.*
The coordinates have been added.
*Line 602: is that for TROP or for both relaxation experiments?*
There is little stratospheric influence in subtropics, so it is probably good to specify that we are talking about tropical teleconnections. The revised text reads:
*"In subtropical regions with high signal-to-noise ratio, as little as ~10 members (or less) are sufficient for the ensemble mean to represent at least 2/3 of the tropical teleconnection signal in SLP."*

*Fig4: Maybe a suggestion in case you reproduce this figure: use transparent shading to improve visibility of the overlapping points.*
Thank you for this suggestion. In the revised version we applied a transparent shading and it somewhat improved visibility, although, because of large amount of dots, it is impossible to fully reveal all overlapping dots.

*Line 560: missing "and"*
This has been added.

*Line 560: missing "and"*
This has been added.